# Stress controls heterochromatin inheritance via histone H3 ubiquitylation

Bharat Bhatt[1], Yi Wei[1,5], Ashis Kumar Pradhan[1], Jothy Dhakshnamoorthy[1], Martin Zofall[1], Hua Xiao[1], Drisya Vijayakumari[1], Shweta Jain[1], Hernan Diego Folco[1], Hongyun Qi[2], David A. Ball[3], Tatiana S. Karpova[3], David Wheeler[1], Jiemin Wong[4] & Shiv I. S. Grewal[1✉]

Heterochromatin, marked by histone H3 lysine 9 methylation, can be epigenetically inherited through cell division[1–3], maintaining gene repression that preserves cell identity and enables adaptation to environmental challenges[2–6]. Studies on *Schizosaccharomyces pombe* have shown that heterochromatin propagation depends on the read–write mechanism, wherein a sufficient density of H3K9me3-modified nucleosomes, stabilized by histone deacetylases, concentrates Clr4[SUV39H] on chromatin to promote further deposition of H3K9 methylation[7–9]. Whether other mechanisms control heterochromatin propagation by means of Clr4[SUV39H], a subunit of the E3 ubiquitin ligase complex ClrC[10–12], was unknown. Here we uncover a ubiquitin-dependent heterochromatin heritability regulatory hub (HRH) that broadly governs heterochromatin propagation, even without histone deacetylase activity. The HRH is tuned by the limiting factor Raf1[DDB2], a substrate receptor for the ClrC ubiquitin ligase. In addition to linking Clr4[SUV39H] to other ClrC components on chromatin, Raf1[DDB2] acts in a dosage-dependent manner to promote ubiquitination of histone H3 at lysine 14 (H3K14ub), which is critical for heterochromatin self-propagation. HRH is intricately linked to environmentally responsive pathways, including nonsense-mediated decay (NMD) and target of rapamycin (TOR) signalling, enabling cells to adapt to changing conditions. By modulating heterochromatin propagation, cells leverage the HRH to gain resistance to antifungal agents and adapt to high temperature. Thus, heterochromatin self-propagation is actively regulated by means of H3K14ub in response to external stimuli, with broad implications for understanding mechanisms governing rapid changes in the epigenetic landscape in physiology and disease.

Heterochromatin, a repressive chromatin state, is crucial for various chromosomal processes in eukaryotes[1]. Heterochromatin suppresses transcription and recombination at repetitive DNA elements and prevents lineage-inappropriate gene expression during development[1,3]. Heterochromatin also enables adaptive, heritable gene repression in response to environmental change without genetic mutation[2,4–6,13,14]. Epigenetic inheritance of heterochromatin occurs through a self-templating mechanism, enabling persistent gene repression[1]. Despite its central role in cell identity and adaptation, the mechanisms underlying stable heterochromatin propagation remain to be fully elucidated.

*Schizosaccharomyces pombe* offers an ideal genetic system for studying heterochromatin propagation. Clr4, an orthologue of *Drosophila* Su(var)3–9 and mammalian SUV39H, methylates histone H3 at lysine 9 (H3K9)[15,16]. H3K9 methylation (H3K9me) enables chromatin association of HP1 proteins and their associated effectors to assemble both discrete facultative heterochromatin islands and major heterochromatin domains found at the silent mating-type (*mat*) region, centromeres and telomeres[15,17]. At the silent *mat* region, heterochromatin is nucleated at the centromere-homologous (*cenH*) element through RNA interference (RNAi)-mediated targeting of Clr4[SUV39H], followed by spreading of H3K9me and HP1 to adjacent sequences[18].

Clr4[SUV39H] binds to pre-existing trimethylated H3K9 (H3K9me3)-marked nucleosomes through its chromodomain ('read') and catalyses further H3K9me deposition ('write')[8]. This read–write mechanism is crucial for the spreading and epigenetic inheritance of heterochromatin[8]. Efficient self-propagation of heterochromatin requires a critical H3K9me3 density to maintain sufficient Clr4[SUV39H] bound to chromatin[7,9]. Histone deacetylases (HDACs), such as the class II HDAC Clr3, help maintain H3K9me3 density[9,19] by blocking access to nucleosome-disrupting chromatin remodellers[20]. Whether other mechanisms control heterochromatin propagation, including those acting at the level of Clr4-containing E3 ubiquitin ligase complex ClrC, has not been explored.

[1]Laboratory of Biochemistry and Molecular Biology, National Cancer Institute, National Institutes of Health, Bethesda, MD, USA. [2]State Key Laboratory of Molecular Biology, Shanghai Key Laboratory of Molecular Andrology, Institute of Biochemistry and Cell Biology, Shanghai Institutes for Biological Sciences, Chinese Academy of Sciences, Shanghai, China. [3]Laboratory of Receptor Biology and Gene Expression, National Cancer Institute, National Institutes of Health, Bethesda, MD, USA. [4]Shanghai Key Laboratory of Regulatory Biology, Institute of Biomedical Sciences and School of Life Sciences, East China Normal University, Shanghai, China. [5]Present address: School of Life Science and Technology, China Pharmaceutical University, Nanjing, China. ✉e-mail: grewals@mail.nih.gov

Here we identify an H3K14ub-dependent heterochromatin HRH operating directly at the level of ClrC. This regulatory hub integrates signals from growth-condition-responsive pathways to govern the spread and epigenetic inheritance of heterochromatin.

## NMD affects heterochromatin propagation

The *S. pombe* ClrC shares structural resemblance to CUL4-DDB1-DDB2 E3 ubiquitin ligase complexes. Beyond the Clr4[SUV39H] methyltransferase, ClrC includes the WD-40 protein Raf1, the Zn finger protein Raf2, the β-propeller protein Rik1 and the cullin family protein Cul4 (refs. 10–12,21,22). Rik1 closely resembles human DDB1, which together with CUL4 participates in histone methylation[23]. Raf1 is similar to human DDB2 and probably a DCAF (DDB1 and CUL4 associated factor), serving as the substrate receptor for E3 ubiquitin ligases[23–25]. Raf1 features two conserved WDxR motifs[23–25] (residues 515–518 and 573–576).

To probe WDxR motif function in Raf1, we generated *raf1[R518H]* and *raf1[R576H]* alleles and assessed heterochromatic silencing at the *mat* region using the sensitive *REIIΔ mat2P::ura4+* reporter (Extended Data Fig. 1a). In *mat1-M* cells lacking the local *REII* silencer, defects in heterochromatin assembly result in derepression of *mat2P* and haploid meiosis, indicated by dark brown staining of cells when exposed to iodine vapour[7,9,26]. By contrast, cells with functional heterochromatic silencing stain yellow. The nearby *mat2P::ura4+* reporter provides an extra readout for heterochromatic silencing when tested on medium lacking uracil (−URA) or counter-selective 5-fluoroorotic acid (FOA) medium. Both *raf1[R518H]* and *raf1[R576H]* showed loss of *mat2P::ura4+* silencing and dark iodine staining (Extended Data Fig. 1a). However, their effects on silencing of *ura4+* inserted at the *cenH* nucleation site (*Kint2::ura4+*) differed. Whereas *Kint2::ura4+* in *raf1[R518H]* showed severe silencing defects with loss of H3K9me3 across the *mat* region, *Kint2::ura4+* in *raf1[R576H]* cells was only weakly affected (Extended Data Fig. 1a,b). Notably, *raf1[R576H]* maintained H3K9me3 around the *cenH* nucleation centre but failed to spread it to surrounding sequences (Extended Data Fig. 1b).

We then investigated how *raf1[R576H]* affects heterochromatin spreading. We performed an unbiased genetic screen for factors that restore silencing of *REIIΔ mat2P::ura4+* in *raf1[R576H]* cells (Fig. 1a). Three suppressors reestablished silencing, indicated by increased growth on FOA medium and light iodine staining. Two mutations mapped to *esl1* (also called *ebs1*), and one to *upf1* (Fig. 1b). Upf1 and Esl1 are components of the NMD pathway[27–29]. Deleting *upf1* or *esl1* reproduced suppression of the *raf1[R576H]* silencing defects (Fig. 1c), confirmed by PCR with reverse transcription (RT–PCR) analysis of the *ura4+* reporter (Fig. 1d). Chromatin immunoprecipitation with sequencing (ChIP–seq) showed increased H3K9me3 and Swi6[HP1] across the silent *mat* region in *raf1[R576H] upf1Δ* and *raf1[R576H] esl1Δ* (Fig. 1e,f), with a similar increase in heterochromatin spreading at subtelomeric regions (Extended Data Fig. 1c). By contrast, centromeric H3K9me3 levels were comparable to wild-type (WT) cells in the single and double mutants, consistent with persistent RNAi-directed ClrC targeting to pericentromeric repeats (Extended Data Fig. 1d). These results show that loss of NMD components can rescue the heterochromatin propagation defect of *raf1[R576H]*.

## NMD controls ClrC subunit Raf1[DDB2] levels

To explain how NMD loss suppresses heterochromatin defects in *raf1[R576H]*, we considered that NMD competes with RNAi for centromeric repeat transcripts, so NMD defects would enhance RNAi-mediated small RNA production and heterochromatin assembly. However, Upf1 loss did not increase small RNA production (Extended Data Fig. 2a). Moreover, the RNAi factor Ago1 was dispensable for restored silencing and H3K9me3 enrichment across the silent *mat* region in *raf1[R576H] upf1Δ* cells (Extended Data Fig. 2b,c). In addition, H3K9me3 levels and heterochromatic silencing were increased in *upf1Δ* cells relative to WT cells, even when the RNAi-dependent heterochromatin nucleation

centre *cenH* was replaced with *ura4+* (*KΔ::ura4+*) (Extended Data Fig. 2d). Thus, the suppression of heterochromatin defects in *raf1[R576H]* cells on loss of NMD components is not due to increased RNAi-mediated heterochromatin assembly.

We asked whether NMD regulates the expression of a critical factor necessary for heterochromatin propagation. RNA sequencing (RNA-seq) showed reduced *raf1* mRNA levels in *raf1[R576H]* compared with WT (Fig. 2a and Extended Data Fig. 2e). *raf1* mRNA levels increased in *raf1[R576H] upf1Δ* cells, exceeding WT levels (Fig. 2a and Extended Data Fig. 2f). This effect was specific to *raf1*, as transcripts encoding other ClrC components were unaffected (Supplementary Table 1). RNA immunoprecipitation (RIP) showed that Upf1 associates with *raf1* mRNA, suggesting NMD directly controls *raf1* expression (Fig. 2b). Indeed, NMD pathway defects increased *raf1* mRNA and Raf1 protein levels (Fig. 2c and Extended Data Fig. 2g). Loss of Upf1 resulted in higher *raf1* expression, not only in *raf1[R576H]* but also in *raf1[WT]* cells (Extended Data Fig. 2g). Therefore, although our genetic screen using the low-expression *raf1[R576H]* mutant allele was pivotal in identifying *raf1* as an NMD target, these analyses show that NMD also suppresses *raf1[WT]* expression. Thus, *raf1* is a bona fide NMD target involved in regulating heterochromatin propagation.

We noticed *raf1* mRNA contains cryptic, inefficiently spliced introns linked to RNA decay[30] (Extended Data Fig. 3a). These introns are detected in *upf1Δ* cells, consistent with the appearance of cryptic introns when target RNA decay pathways are disabled[30] (Extended Data Fig. 3a). Our analyses also revealed that, like NMD, splicing machinery affects *raf1* mRNA. From the screen that initially implicated NMD in *raf1* regulation (Fig. 1a), we characterized two more mutants: one carried a mutation in *upf2*, which also functions in NMD, and the other carried a mutation in *sap49*, which encodes a U2 snRNP–associated RNA-binding splicing factor. Introducing the *sap49[A175V]* mutation in *raf1[R576H]* cells restored heterochromatic silencing and H3K9me3 spreading at the silent *mat* region (Extended Data Fig. 3b,c) and partially restored H3K9me3 spreading at subtelomeres (Extended Data Fig. 3d). No major changes in H3K9me3 were observed at centromeres in which RNAi persistently targets heterochromatin (Extended Data Fig. 3e). The *sap49[A175V]* markedly upregulated *raf1* mRNA and Raf1 protein, mirroring *upf1Δ* (Extended Data Fig. 3f,g). Together with previous links between splicing and NMD[31,32], these results indicate that Sap49 may promote *raf1* mRNA decay through NMD.

## Raf1 drives heterochromatin propagation

To determine whether NMD loss restores heterochromatin propagation simply by increasing Raf1 levels, we ectopically overexpressed *raf1[R576H]* under an inducible promoter (*raf1[R576H]-oe*) and, for comparison, overexpressed the *raf1[WT]* (*raf1-oe*). Raf1 expression at levels comparable to those in *upf1Δ* cells restored silencing in *raf1[R576H]* cells (Extended Data Fig. 3h,i). *raf1[R576H]-oe* rescued silencing to the same extent as *raf1[WT]-oe* (Fig. 2d). Restoration of silencing was accompanied by H3K9me3 spreading across the silent *mat* region (Fig. 2e), and H3K9me3 levels were comparable between *raf1[R576H]-oe* and *raf1[WT]-oe* cells. These findings indicate that the *R576H* mutation does not impair Raf1 function; rather, heterochromatin defects in *raf1[R576H]* cells reflect reduced Raf1 abundance. *raf1[R576H]-oe* also rescued subtelomeric heterochromatin spreading defects in *raf1[R576H]* cells (Fig. 2f). Thus, NMD-mediated control of Raf1 abundance is a key control point in the regulation of heterochromatin propagation.

## Raf1[DDB2] abundance governs ClrC binding

We then investigated how Raf1 abundance affects H3K9me3 and heterochromatin propagation. Our analyses suggested that Raf1 is limiting for ClrC assembly, particularly in recruiting Clr4[SUV39H], among other functions (below). Glycerol-gradient fractionation revealed a substantial

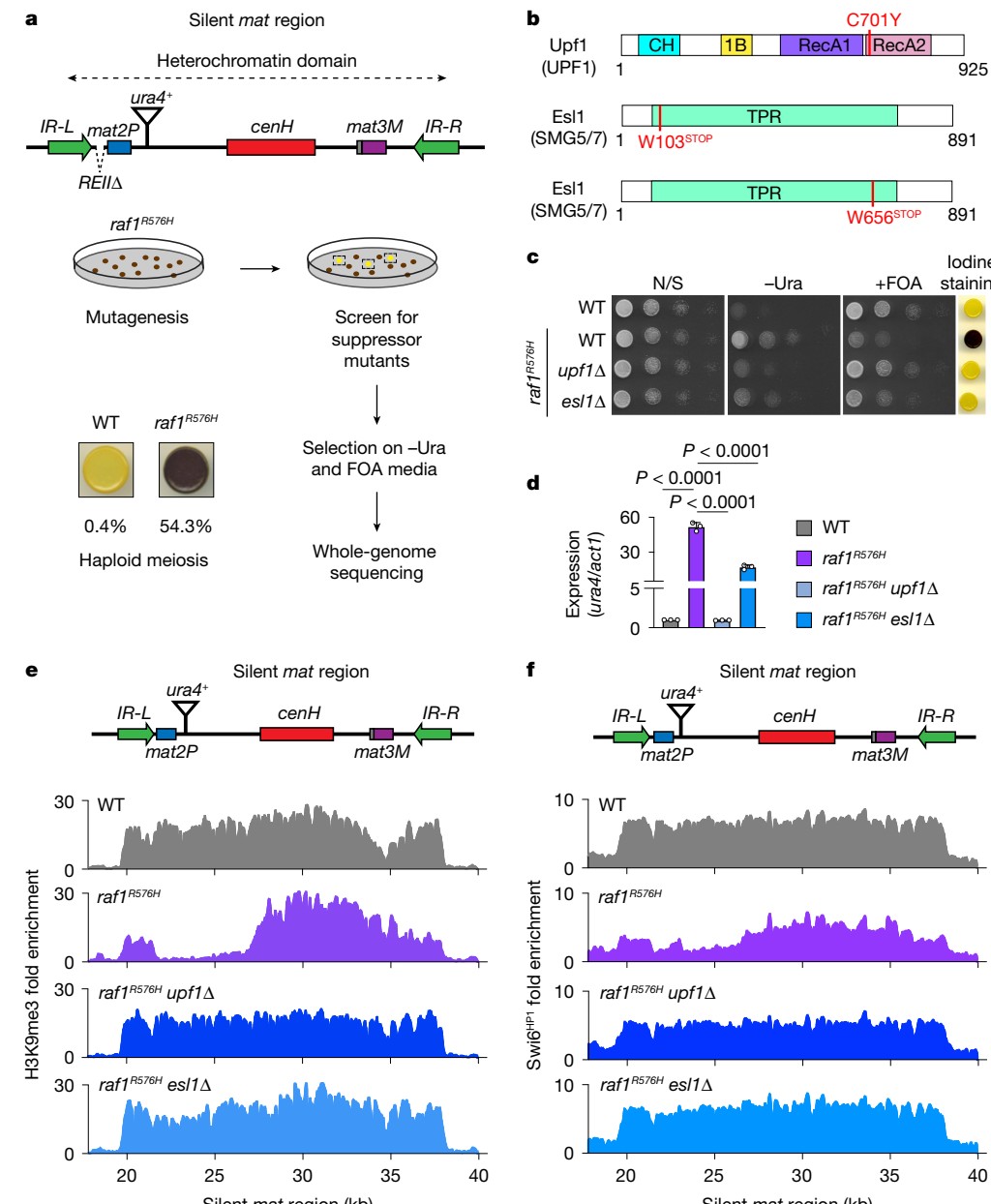

**Fig. 1 | Loss of NMD components rescues heterochromatin spreading defects.**
**a**, Genetic screen to identify suppressor mutants that restore silencing at the *mat* region in the *raf1*[R576H] mutant. Staining of colonies with iodine vapours as well as growth on uracil-deficient and FOA-containing media were used to assess rescue of silencing defects. Percentage of cells undergoing haploid meiosis is indicated. **b**, Domain organization of Upf1 and Esl1 showing the mutations (in red) identified in the suppressor screen. **c**, Restoration of heterochromatic silencing on loss of NMD components in *raf1*[R576H] mutant cells. Serial dilutions were plated on non-selective, uracil-deficient and FOA-containing media to assess *mat2P::ura4*+ expression. Haploid meiosis was assessed by iodine staining. **d**, RT–qPCR analysis of *ura4*+ expression (data are presented as mean ± s.d.; $n = 3$ independent experiments, adjusted $P$ value by one-way analysis of variance followed by Holm–Sidak multiple comparisons test). **e**,**f**, ChIP–seq analysis of H3K9me3 (**e**) and Swi6[HP1] (**f**) distribution at the silent *mat* region in the indicated strains. Data are representative of two independent experiments. +FOA, FOA-containing; N/S, non-selective; –Ura, uracil-deficient.

fraction of monomeric Clr4[SUV39H], indicating a weak association with other ClrC components (Fig. 3a). On Raf1 overexpression, Clr4[SUV39H] shifted to higher molecular weight fractions (Fig. 3a). This shift correlated with increased incorporation of Clr4[SUV39H] into ClrC, evidenced by enhanced Clr4[SUV39H] association with the ClrC subunit Raf2 in both *raf1-oe* and *upf1Δ* cells (Fig. 3b and Extended Data Fig. 4a). Therefore, Raf1 levels probably govern the association of Clr4[SUV39H] with other ClrC components.

The Raf1 WD-repeat domain adopts a β-propeller fold that mediates protein interactions[33]. We wondered whether Raf1 directly engages Clr4[SUV39H]. The Raf1 WD-repeat region bound Clr4[SUV39H] in vitro (Extended

Data Fig. 4b). In vivo, Clr4[SUV39H] association with Raf2 correlated with Raf1 abundance. Clr4 failed to associate with Raf2 in cells lacking Raf1, and only weak association was detected in *raf1*[R576H] cells with low Raf1 levels (Fig. 3c). By contrast, Clr4–Raf2 association increased with higher Raf1 levels in cells carrying the *raf1*[WT] allele, and Raf1 overexpression produced a marked enhancement of Clr4 association with Raf2. These findings further suggest that Raf1 is a limiting factor for incorporation of Clr4[SUV39H] into ClrC.

To determine whether Raf1 levels influence the chromatin-bound state of ClrC, we tracked Raf2 dynamics in live cells. Raf2 tagged with HaloTag (Raf2-Halo) formed two to three foci, corresponding to

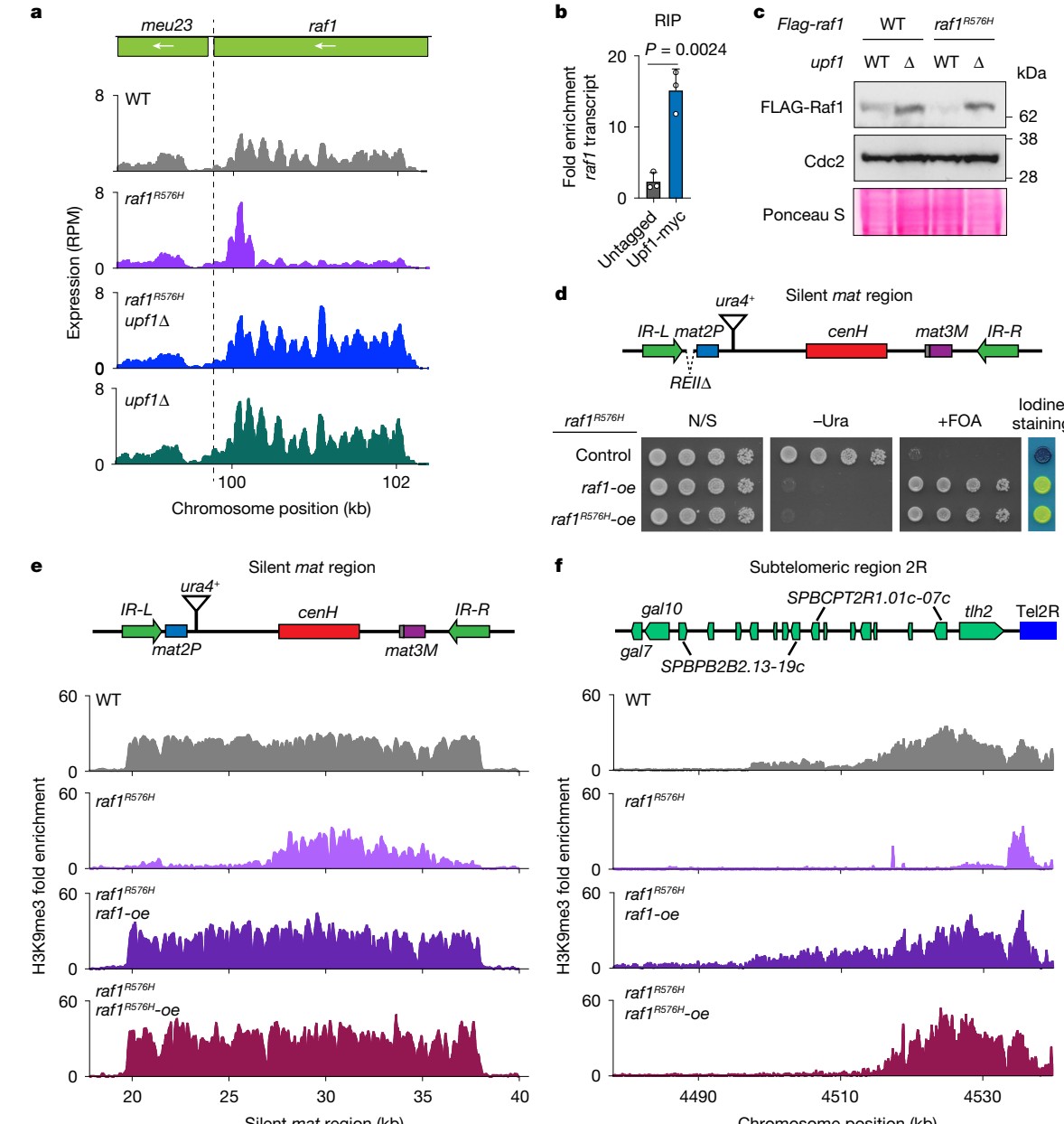

**Fig. 2 | The NMD machinery regulates the expression of the ClrC subunit Raf1[DDB2]. a**, RNA-seq expression profile of *raf1* and a neighbouring gene in the indicated strains. **b**, RIP to detect Upf1 association with *raf1* transcript. Data normalized to *act1* (data are presented as mean ± s.d.; *n* = 3 independent experiments, *P* value by two-tailed unpaired *t*-test). **c**, Western blot analysis of Raf1 in the indicated strains. Cdc2 and Ponceau S serve as loading controls. **d**, Analysis of heterochromatic silencing in *raf1[R576H]* mutant cells on

overexpression of WT (*raf1-oe*) or mutant (*raf1[R576H]-oe*) *raf1* under the control of *nmt1* promoter. Serial dilution monitored *mat2P::ura4+* expression, and iodine staining assessed haploid meiosis. **e**,**f**, ChIP–seq analysis of H3K9me3 distribution across the silent *mat* region (**e**) and subtelomeric region of Chromosome 2R (**f**) in the indicated strains. Data are representative of two independent experiments. Raw western blots are presented in Supplementary Fig. 1. RPM, reads per million mapped reads.

clustered heterochromatic loci at the nuclear periphery (Extended Data Fig. 4c). Single-molecule tracking (SMT) revealed slow and fast diffusing Raf2-Halo particles and their diffusion coefficients (*D*) (Extended Data Fig. 4d and Supplementary Videos 1 and 2). As a chromatin-bound control, histone H2B tagged with Halo (H2B-Halo) showed two similar diffusive states. On Raf1 overexpression, the slow-diffusing Raf2 population increased, indicating a redistribution of ClrC to a less mobile, chromatin-bound state. Moreover, the survival distribution indicated reduced dissociation of chromatin-bound ClrC on overexpression of Raf1 (Extended Data Fig. 4e).

To further analyse ClrC mobility, we classified Raf2 trajectories into different diffusive states using a systems-level classification algorithm

perturbation-expectation maximization (pEM) (Extended Data Fig. 4f). Three subdiffusive Raf2 mobility states were identified (Extended Data Fig. 4f). State 3 showed a radius of confinement ($R_c$) corresponding to the average nuclear radius, consistent with the free state that was not observed for H2B. States 1 and 2 had smaller $R_c$ values and were considered confined. Notably, the *D* and $R_c$ of Raf2-Halo confined states 1 and 2 were similar to the two H2B states, indicating that they are the chromatin-bound states (Extended Data Fig. 4g,h). Raf1 overexpression led to an increased proportion of the Raf2 confined states 1 and 2 and a reduction in the free state (Extended Data Fig. 4i), indicating a shift of Raf2 molecules towards the chromatin-associated population. Together, these results indicate that higher levels of Raf1 not only

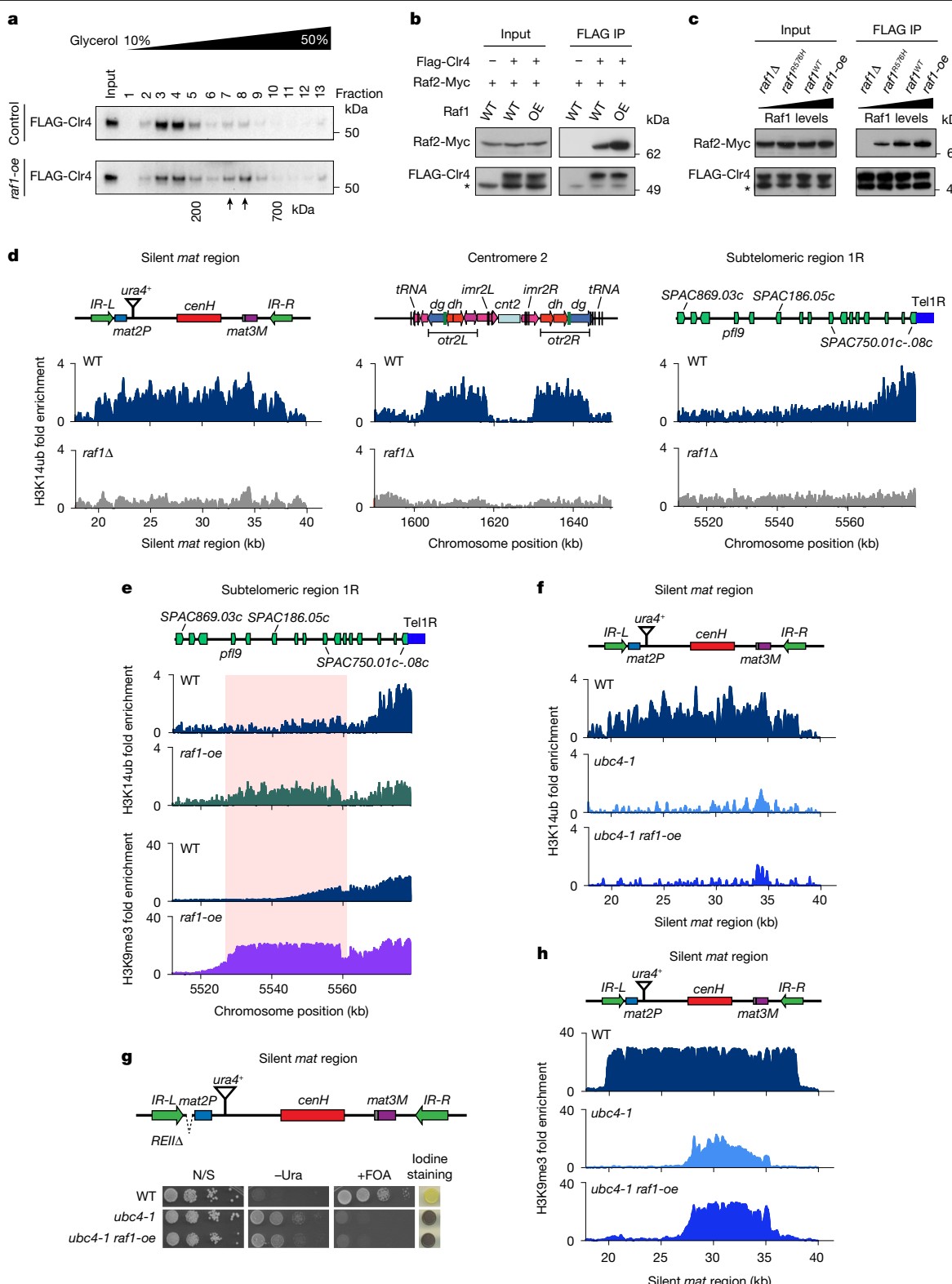

**Fig. 3 | Raf1[DDB2] connects Clr4[SUV39H] to ClrC and mediates H3K14ub to promote heterochromatin spreading. a**, Glycerol-gradient fractionation showing Clr4[SUV39H] distribution from low to high molecular weight fractions on overexpression of *raf1* (*raf1-oe*) under the control of *nmt1* promoter. **b**, Co-immunoprecipitation of Raf2 with Clr4[SUV39H] in the indicated strains. OE, overexpression. **c**, Co-immunoprecipitation showing Raf2 and Clr4[SUV39H] association in cells expressing different levels of Raf1. The asterisk * denotes the non-specific band. **d**, ChIP–seq analysis of H3K14ub distribution across the silent *mat* region, centromere 2 and subtelomeric region 1R in WT and *raf1Δ*

cells. **e**, ChIP–seq analysis of H3K14ub and H3K9me3 distribution across subtelomeric region 1R in the indicated strains. **f**, ChIP–seq analysis of H3K14ub distribution across the silent *mat* region in the indicated strains. **g**, Heterochromatic silencing in *ubc4-1* cells on overexpression of *raf1* (*raf1-oe*) under the control of *adh1* promoter. Serial dilutions on indicated media were used to assess *mat2P::ura4*[+] expression. Haploid meiosis was evaluated by iodine staining. **h**, ChIP–seq analysis of H3K9me3 distribution across the silent *mat* region in the indicated strains. All data are representative of two independent experiments. IP, immunoprecipitation.

promote ClrC integrity but also increase the proportion of ClrC bound to chromatin.

The enhanced chromatin association of ClrC observed with increased Raf1 levels was also evident from direct visualization of Raf2 localization. The number and intensity of Raf2-Halo foci increased considerably in *raf1-oe* and *upf1Δ* cells, both of which contain higher Raf1 levels (Extended Data Fig. 5a,b). In *raf1-oe* cells, higher Raf2 foci intensity correlated with enhanced Raf2 spreading, especially at subtelomeric regions lacking defined heterochromatin boundaries (Extended Data Fig. 5c), whereas Raf2 distribution at heterochromatic regions flanked by boundary elements, such as the silent *mat* region, remained unchanged (Extended Data Fig. 5d). Raf1 overexpression also increased ClrC binding at heterochromatin islands, including genes responsive to developmental or environmental signals, such as *pho1* (Extended Data Fig. 5e).

Collectively, these analyses suggest that elevated Raf1 stabilizes and prolongs ClrC chromatin association, concentrating this histone methyltransferase and ubiquitin ligase complex, which ultimately promotes the read–write activity that drives heterochromatin propagation.

## Raf1[DDB2] DCAF directs H3K14ub

ClrC monoubiquitylates histone H3 at lysine 14 (H3K14ub), which enhances Clr4[SUV39H] methyltransferase activity in vitro[34–36], however, its in vivo relevance remained unclear. We asked whether histone H3 is the physiological substrate of the ClrC E3 ligase and whether Raf1 promotes H3K14ub beyond its dose-dependent effects on ClrC chromatin association.

We profiled H3K14ub genome-wide by ChIP–seq with an anti-H3K14ub antibody[37]. H3K14ub was enriched across major heterochromatin domains, including the silent *mat* region, pericentromeres and subtelomeres, but was completely abolished in cells lacking Raf1 (Fig. 3d). This result indicates that H3K14ub is a prominent heterochromatin mark whose deposition requires the DCAF Raf1. Raf1 overexpression markedly increased H3K14ub, particularly at subtelomeres that lack boundary elements, and this increase correlated with enhanced H3K9me3 spreading (Fig. 3e). Thus, Raf1 dosage is critical for H3K14ub and H3K9me3, and merely increasing Raf1 enhances heterochromatin propagation.

To test the requirement for H3K14ub in heterochromatin propagation, we examined the E2 ubiquitin-conjugating enzyme Ubc4 mutant *ubc4-1* (ref. 38). H3K14ub was markedly reduced across heterochromatin domains, including the *mat* region (Fig. 3f). Because Ubc4 is not a core component of the dually functioning ClrC, which mediates both ubiquitylation and H3K9 methylation, this mutation isolates the contribution of H3K14ub to heterochromatin assembly. In *ubc4-1*, loss of H3K14ub correlated with impaired heterochromatin spreading and silencing at the silent *mat* region (Fig. 3g,h). Although H3K9me3 was established at the *cenH* nucleation site, where the RNAi machinery recruits ClrC, it failed to spread in the H3K14ub-deficient *ubc4-1* mutant (Fig. 3h). Raf1 overexpression did not restore H3K14ub, H3K9me3 spreading or heterochromatic silencing in *ubc4-1* cells (Fig. 3f,g,h).

H3K14ub stimulates Clr4[SUV39H] methyltransferase activity in vitro[34–36] and is therefore expected to hyperactivate Clr4[SUV39H], leading to H3K9me3 accumulation that stimulates the read–write mechanism of heterochromatin propagation. Accordingly, *ubc4-1* probably disrupts the H3K14ub and Clr4[SUV39H] crosstalk, reducing H3K9me3 below the threshold required for efficient heterochromatin propagation[1,7].

## H3K14ub defines a new regulatory pathway

Clr3, the HDAC that deacetylates H3K14, is essential for heterochromatin propagation[9,19]. By suppressing histone turnover, Clr3 maintains sufficient H3K9me3 density required for Clr4 read–write activity[1]. Loss of Clr3 reduces H3K9me3 density and impairs heterochromatin propagation and silencing[9,19] (Fig. 4a). We also found reduced H3K14ub in *clr3Δ* cells, particularly across regions surrounding the *cenH* nucleation site (Fig. 4b), prompting us to test whether elevating Raf1 could restore heterochromatin assembly.

Increasing Raf1, by means of *raf1-oe* or *upf1Δ*, reinstated heterochromatic silencing of the *REIIΔ mat2P::ura4+* reporter in *clr3Δ* cells (Fig. 4a) and correlated with increased H3K14ub levels across the silent *mat* region (Fig. 4b). Whereas *clr3Δ* cells showed H3K9me3 enrichment mainly near *cenH*, H3K9me3 spread across the entire domain in *clr3Δ raf1-oe* and *clr3Δ upf1Δ* cells (Fig. 4c). This suggests a parallel mechanism such that boosting Raf1 levels, which enhances ClrC chromatin association and H3K14ub, bypasses the requirement for the Clr3 HDAC activity in heterochromatin propagation (Fig. 4d).

Raf1 upregulation in *clr3Δ* cells phenocopies loss of the anti-silencing factor Epe1 (ref. 39). Because Epe1 undergoes ubiquitin-dependent proteolysis[40,41], we asked whether Raf1 alters Epe1 abundance. Epe1-GFP (green fluorescent protein) foci number and intensity were similar in WT and *raf1-oe* cells (Extended Data Fig. 6a), indicating Raf1 does not act by reducing Epe1 levels. Genetic analyses showed that Epe1 and Raf1 act independently to suppress the silencing defect of Clr3 catalytic mutant (*clr3[D232N]*) cells. Deleting *epe1* or *upf1* each partially restored *REIIΔ mat2P::GFP* reporter silencing, and the double deletion produced an additive effect and near-complete rescue (Extended Data Fig. 6b). Together with the unchanged Epe1 levels on Raf1 overexpression, these results support a distinct Raf1-mediated pathway controlling heterochromatin propagation.

To further validate a Raf1-based mechanism for controlling heterochromatin propagation, we tested whether increasing Raf1 mitigates defects in other contexts. In cells carrying a mutation in one of the three histone H3 copies (*hht2[G13D]*), the reduction in H3K9me3 density impairs heterochromatin propagation[7]. Notably, introducing *raf1-oe* or *upf1Δ* into *hht2[G13D]* cells restored *REIIΔ mat2P::ura4+* silencing (Extended Data Fig. 7a). *hht2[G13D]* cells also showed defective H3K14ub across the silent *mat* region (Extended Data Fig. 7b), a defect rescued by increased Raf1 levels in *upf1Δ* cells and accompanied by restored H3K9me3 spreading across the domain (Extended Data Fig. 7c,d).

We further explored whether elevating Raf1 bypasses other requirements for heterochromatin propagation. Indeed, *raf1-oe* or *upf1Δ* restored silencing in cells lacking the FACT (facilitates chromatin transcription) subunit Pob3, the SMARCAD1 SNF2 remodeller Fft3 or the nuclear rim protein Amo1[NUPL2] (Extended Data Fig. 7e), factors that preserve H3K9me3 density[26,42,43]. Collectively, these results identify Raf1-mediated control of ClrC chromatin association and H3K14ub as a previously unrecognized mechanism supporting heterochromatin maintenance and propagation.

## Raf1 levels control self-propagation

We tested whether Raf1 abundance influences sequence-independent heterochromatin inheritance. Heterochromatin was nucleated at an ectopic site by reversibly tethering TetR-Clr4 (Clr4[SUV39H] fused to the TetR DNA-binding domain) to six tetracycline operators located upstream of the *ade6+* reporter (Fig. 4e). Without tetracycline (*tetR-clr4[ON]*), TetR-Clr4 induced H3K9me3 and silenced *ade6+* (red colonies). On release of TetR-Clr4 (*tetR-clr4[OFF]*), H3K9me3 and *ade6+* silencing were rapidly lost, probably because the local chromatin concentration of endogenous Clr4[SUV39H] was insufficient to sustain read–write activity and maintain ectopic heterochromatin[9,26,44,45]. By contrast, releasing TetR-Clr4 in *upf1Δ* cells with elevated Raf1 levels allowed *ade6+* silencing and H3K9me3 at the ectopic site to persist for several generations (Fig. 4e), indicating that increased Raf1 abundance, which enhances ClrC chromatin association, is sufficient to support heterochromatin self-propagation.

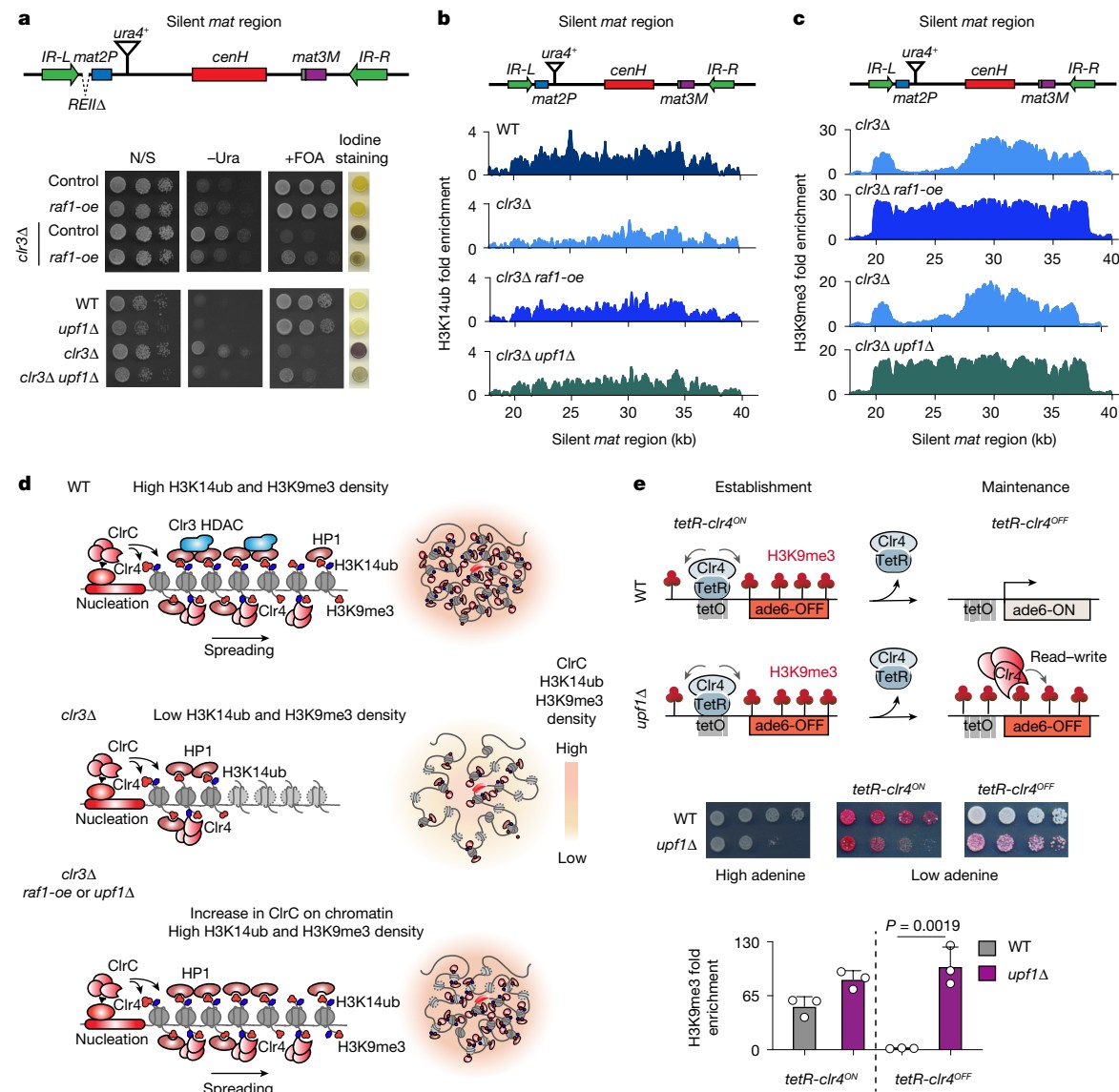

**Fig. 4 | Raf1^DDB2 DCAF-directed H3K14ub reveals a new pathway for heterochromatin propagation. a**, Rescue of silencing defects in *clr3Δ* cells by *raf1* overexpression (*raf1-oe*) from the *adh1* promoter or by deletion of *upf1* (*upf1Δ*). Serial dilution assessed *mat2P::ura4*+ expression, and iodine staining assessed haploid meiosis. **b,c**, ChIP–seq profiles of H3K14ub (**b**) and H3K9me3 (**c**) across the silent *mat* region in the indicated strains. **d**, Schematic illustrating restoration of heterochromatin propagation in *clr3Δ* cells on increased Raf1 expression through *raf1-oe* or *upf1Δ*. ClrC, recruited to the nucleation site, deposits H3K9me3 and H3K14ub. In WT cells, the Clr3 HDAC suppresses histone turnover, preserving H3K14ub, H3K9me3 and associated Clr4^SUV39H, to support heterochromatin spreading through the read–write mechanism (top). Loss of Clr3 lowers H3K9me3 and H3K14ub density, compromising Clr4^SUV39H read–write activity and heterochromatin propagation (middle). Elevated Raf1 restores Clr4/ClrC, H3K14ub and H3K9me3 density, enabling heterochromatin propagation even in the absence of Clr3 (bottom). **e**, Schematic of ectopic heterochromatin establishment and maintenance in the indicated strains (top). A TetR-Clr4 fusion, expressed from a thiamine-repressible promoter (*tetR-clr4^ON*), nucleates heterochromatin at a *tetO-ade6*+ reporter. Heterochromatin inheritance by read–write activity of endogenous Clr4^SUV39H was tested after TetR-Clr4 release in the presence of thiamine (*tetR-clr4^OFF*). Red colonies on low-adenine medium indicate *ade6*+ repression, whereas white colonies indicate expression (middle). ChIP analysis of H3K9me3 at the *ade6*+ reporter in the indicated strains (bottom). Data are mean ± s.d. (*n* = 3 independent experiments); *P* values by two-tailed unpaired *t*-test. All data are representative of at least two independent experiments.

## Raf1 abundance is regulated in adaptation

NMD-regulated Raf1 levels substantially affect heterochromatin propagation, suggesting this mechanism operates under physiological conditions. Self-propagation of heterochromatin enables heritable gene reprogramming and environmental adaptation[2,4,6,13,14]. A recent study demonstrated that cells acquire resistance to caffeine through heterochromatin propagation[14]. Because caffeine attenuates NMD activity in other systems[46,47], we investigated whether caffeine treatment affects NMD in *S. pombe*. NMD-regulated transcripts[29]

were upregulated in the presence of caffeine (Extended Data Fig. 8a). Moreover, the *raf1^RS76H* mutant transcript was derepressed, as in *upf1Δ* (Fig. 2a), with a corresponding increase in protein levels (Extended Data Fig. 8b,c). Caffeine treatment also markedly increased WT Raf1 levels (Fig. 5a).

We asked if Raf1 abundance alone favours the development of caffeine resistance. Raf1 overexpression correlated with increased caffeine-resistant colonies (Fig. 5b) and resistance to the antifungal agents, fluconazole and clotrimazole (Fig. 5c). Similar resistance was observed in *upf1Δ* cells (Fig. 5c). By contrast, Raf1 overexpression

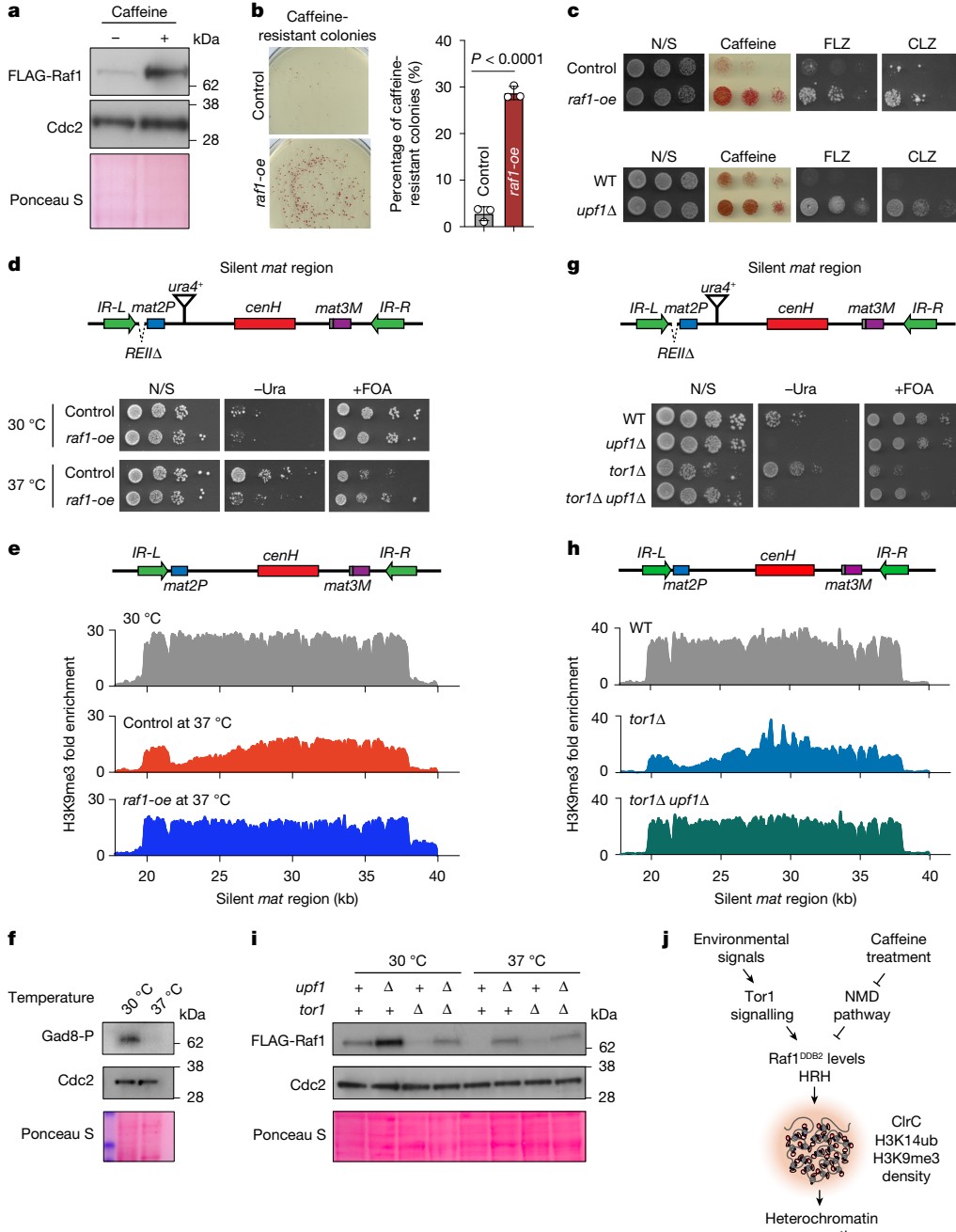

**Fig. 5 | Raf1^DDB2 levels are naturally regulated during heterochromatin adaptation. a**, Western blot of Raf1 in cells treated with 16 mM caffeine for 2 weeks. Cdc2 and Ponceau S serve as loading controls. **b**, Cells with or without *raf1* overexpressed from the *adh1* promoter (*raf1-oe*) were plated on caffeine and caffeine-resistant colonies were quantified (mean ± s.d.; *n* = 3 independent experiments; *P* values by two-tailed unpaired *t*-test). **c**, Caffeine, fluconazole (FLZ) and clotrimazole (CLZ) resistance in *raf1-oe* and *upf1Δ* cells assessed by serial dilutions. **d**,**e**, Heterochromatin assembly and silencing at 30 °C and 37 °C with or without *raf1-oe* assessed by serial dilution analysis of *mat2P::ura4^+* expression (**d**) and by H3K9me3 ChIP–seq across the silent *mat* region (**e**). **f**, Western blot analysis of Gad8 S546 phosphorylation at 30 °C and 37 °C. Cdc2 and Ponceau S serve as loading controls. **g**,**h**, Serial dilution analysis of

*mat2P::ura4^+* expression in *upf1Δ* and *tor1Δ* single and double mutant cells (**g**) and H3K9me3 ChIP–seq across the silent *mat* region in the indicated strains (**h**). **i**, Western blot of Raf1 at 30 °C and 37 °C in the indicated strains. Cdc2 and Ponceau S serve as loading controls. **j**, Model showing the environmental cues converge on the heterochromatin HRH, where Raf1^DDB2 abundance, regulated by TOR and the NMD pathway, modulates heterochromatin propagation to reprogram gene expression and enable adaptation. Raf1^DDB2, a limiting factor for ClrC chromatin association, also mediates H3K14ub, which together maintain high local H3K9me3 density and associated Clr4^SUV39H, critical for heterochromatin spreading and inheritance. **e**,**f**,**i**, Cells were grown at 37 °C for 16 h. Data are representative of two independent experiments.

failed to confer caffeine resistance in cells lacking Clr4^SUV39H, suggesting a heterochromatin-dependent mechanism (Extended Data Fig. 8d,e). ChIP–seq of H3K9me3 in *raf1-oe* caffeine-resistant cells revealed increased H3K9me3 and spreading at known heterochromatin islands and at new genomic locations (Extended Data Fig. 8f),

including loci whose silencing confers caffeine resistance, such as *cup1* implicated in mitochondrial dysfunction[14] (Extended Data Fig. 8g). Thus, Raf1 upregulation is part of a natural mechanism to enhance heterochromatin stabilization, enabling cells to adapt to unfavourable conditions.

## TOR-Raf1 node modulates heterochromatin

We explored whether other conditions affect Raf1 levels. Culturing cells at an elevated temperature (37 °C) reduced Raf1 levels (Extended Data Fig. 9a), which is intriguing given the temperature-dependent alterations in heterochromatin observed in various systems[48–51]. This Raf1 reduction at 37 °C resulted in impaired heterochromatin spreading and disruption in gene silencing (Fig. 5d,e). Raf1 overexpression restored both silencing and H3K9me3 distribution across the silent *mat* interval (Fig. 5d,e), suggesting that cells downregulate Raf1 in response to elevated temperatures, thereby affecting heterochromatin propagation.

We next examined signalling pathways that modulate Raf1. Loss of Tor1, a TORC2 complex subunit[52], severely reduced Raf1 protein and transcript levels (Extended Data Fig. 9b,c). By contrast, Raf1 levels were not affected by mutations in other kinases, including Tor2, the TORC1 subunit that targets RNA processing machinery for H3K9me at meiotic genes but is dispensable for heterochromatin assembly elsewhere[53] (Extended Data Fig. 9b). To test temperature effects on Tor1 activity, we examined Gad8 (human AKT orthologue)[54,55] phosphorylation at 30 °C and 37 °C. Gad8 phosphorylation was high at 30 °C, but undetectable at 37 °C (Fig. 5f). Cells lacking Gad8 also showed reduced Raf1, similar to *tor1Δ* (Extended Data Fig. 9b). These findings suggest that high temperature inactivates the Tor1 signalling cascade, which is required for proper Raf1 expression. The reduction in Raf1 levels in both *tor1Δ* and *gad8Δ* cells also resulted in a decrease in caffeine-resistant colonies (Extended Data Fig. 9d,e), a phenotype reversed by Raf1 overexpression (Extended Data Fig. 9f).

On the basis of our results, we hypothesized that reduced Raf1 abundance drives the heterochromatin defects previously reported in *tor1Δ* cells[56]. Indeed, increasing Raf1 in *upf1Δ* suppressed heterochromatin propagation and silencing defects in a *tor1Δ* background (Fig. 5g,h). However, Raf1 levels were lower in the *upf1Δ tor1Δ* double mutant than in *upf1Δ* alone (Fig. 5i), indicating that Tor1 affects Raf1 independently of NMD. Supporting this, inactivating Tor1 by culturing *upf1Δ* cells at 37 °C reduced Raf1 expression compared with 30 °C (Fig. 5i). As expected, shifting *upf1Δ tor1Δ* cells from 30 °C to 37 °C caused no further decrease (Fig. 5i). Thus, Upf1 and Tor1 operate through separate pathways to control Raf1 abundance. These findings support a model in which many inputs converge on the control of Raf1 levels, positioning Raf1 as a key component of a regulatory hub controlling heterochromatin robustness and heritability (Fig. 5j).

## Preserving heterochromatin under stress

Given the pivotal role of Raf1-based HRH in heterochromatin propagation, we wondered whether factors influencing Raf1 expression affect epigenetic inheritance of heterochromatin. Because high temperature considerably reduces Raf1 levels, we proposed that this would disrupt the self-propagation of an ectopic heterochromatin domain. We tested this using a robust reporter containing one or two copies of Clr3-attracting sequences (*CAS*) inserted adjacent to the *tetO-ade6*+ gene. *CAS* recruits the Clr3 HDAC, facilitating efficient heterochromatin propagation[9].

We established heterochromatin by tethering TetR-Clr4 (*tetR-clr4*ON) to the *CAS-tetO-ade6*+ reporter in cells cultured at 30 °C, then assessed propagation of the silenced heterochromatic state after TetR-Clr4 release (*tetR-clr4*OFF) at 30 °C or 37 °C. At 30 °C, the heterochromatic state was epigenetically inherited, with propagation efficiency increasing with the number of *CAS* elements, such that two copies of *CAS3* offered the most efficient heterochromatin maintenance (Extended Data Fig. 10a). Conversely, cells cultured at 37 °C failed to maintain the silenced state (Extended Data Fig. 10a).

We then asked whether increasing Raf1 rescues the high-temperature defect in heterochromatin propagation. Using the *2xCAS3-tetO-ade6*+ reporter, we compared the propagation of the ectopic heterochromatin domain in WT and *upf1Δ* cells. After releasing TetR-Clr4, *upf1Δ* cells propagated silenced *ade6*+ even at 37 °C, whereas WT maintained inheritance of the silenced state only at 30 °C (Extended Data Fig. 10b). ChIP revealed greater preservation of H3K9me3 in *upf1Δ* than WT (Extended Data Fig. 10b).

These findings establish that the control of Raf1 levels under different growth conditions represents a bona fide heterochromatin HRH. This hub modulates heterochromatin stability by governing ClrC chromatin association and H3K14ub levels, and is exploited during adaptation to environmental challenges.

## Discussion

The remarkable ability of heterochromatin to be epigenetically inherited across many cell divisions stabilizes gene repression, maintaining committed cell states and enabling adaptation without genetic changes[2–6,14,51]. We have previously shown that histone deacetylation by HDACs suppresses histone turnover, thereby maintaining H3K9me3 density required for heterochromatin propagation through the Clr4SUV39H read–write mechanism[1,7–9]. Here, we identify a previously unrecognized regulatory hub for heterochromatin heritability, the HRH, that responds to environmental signals to control heterochromatin self-propagation.

Given the central role of heterochromatin in epigenetic gene regulation, our discovery of the HRH, with Raf1 DCAF as a key control point, marks a major advance. NMD and TOR signalling, which respond to environmental and developmental cues[27,28,52,57], converge on Raf1, whose abundance ultimately dictates the robustness of heterochromatin propagation. How does elevated Raf1 promote heterochromatin propagation? Raf1 is a limiting factor that links Clr4SUV39H to other ClrC components and facilitates their chromatin association, which alone may create the high local concentration of the H3K9me3 needed to sustain heterochromatin propagation. However, Raf1 DCAF also acts in a dose-dependent manner to promote H3K14ub, a modification required for propagation of heterochromatic structures. This is important because in vitro studies showed that H3K14ub notably increases Clr4SUV39H activity by inducing a conformational shift, enhancing affinity for the histone substrate[34–36], and also provides a binding site for chromodomain-containing amino-terminal region of Clr4/SUV39H[34,37]. Therefore, Raf1 promotes heterochromatin propagation by directly influencing ClrC chromatin association and mediating H3K14ub, which are both essential for achieving the critical H3K9me3 threshold and supporting the read–write activity of Clr4SUV39H. As a DCAF, Raf1 may also target further substrates for ClrC-mediated ubiquitination, including factors that destabilize heterochromatin. Consistently, increasing Raf1 abundance bypasses the requirement for the Clr3 HDAC, mirroring effects observed when histone acetyltransferases or SWI/SNF remodelling enzymes that disrupt nucleosomes are absent[9,20].

Two distinct pathways, HDACs and HRH, maintain the high H3K9me and Clr4SUV39H density needed for read–write propagation of heterochromatin. Under normal growth conditions, cells with basal Raf1 levels probably rely more on HDACs, which safeguard H3K9me3 nucleosomes by blocking access to remodellers and enhance Clr4SUV39H activity[9,20], thereby promoting heterochromatin propagation. This pathway is most effective at loci with high HDAC concentrations[9]. By contrast, the HRH becomes critical under stress or other specific conditions that demand broader reinforcement of heterochromatin heritability. To this end, elevating Raf1 alone is sufficient to promote heterochromatin self-propagation at an ectopic site, even under stress.

The HRH pathway enhances heterochromatin propagation, enabling resistance to caffeine and to the antifungal agents, fluconazole

and clotrimazole. Caffeine increases Raf1 levels (this study) and has also been reported to reduce levels of the negative heterochromatin regulator, Epe1 (ref. 41). We show that these factors operate independently. Raf1 overexpression alone increases caffeine resistance without affecting Epe1 levels. HRH also responds to other stresses. High temperature deactivates TORC2, lowers Raf1 and impairs heterochromatin propagation. Further stress-responsive pathways may also signal to HRH to protect against environmental stressors.

Given the emerging role of epigenetic mechanisms in phenotypic plasticity across diverse systems[58], our discovery that HRH governs stimulus-responsive heterochromatin self-propagation has important implications. Raf1 and ClrC are present in pathogenic fungi and may drive antifungal drug resistance by altering heterochromatin heritability. H3K14ub, a key HRH mark, is conserved in mammalian heterochromatin[37], which undergoes pronounced H3K9me3 changes during development and in response to stress or oncogenic transformation[3,59]. Similar pathways probably operate in other species, including plants, in which ubiquitin ligases enable adaptation to temperature extremes, drought and pathogens[60].

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

## Methods

### Strains and media

Standard methods were used for the culture, sporulation, genetic crossing and manipulation of *S. pombe*. Strains were generated through transformations or genetic crosses followed by tetrad dissection. Strains used in this study are listed in Supplementary Table 2. Gene deletion strains were obtained from Bioneer haploid deletion library version 4.0, except for *tor1* and *gad8*, which were deleted using pFA6a-natMX6 construct. Most epitope-tagged strains were generated through PCR module-based methods. *GFP* or *GFP-raf1* was cloned in the *pDUAL-Pnmt1* or *pUC119-Padh1* vector using In-Fusion cloning. Pombe Minimal Glutamate medium supplemented with adenine, uracil, leucine, histidine and lysine (PMG-5S) was used to overexpress protein under the *nmt1* promoter. All other experiments were performed in yeast extract-rich medium supplemented with adenine (YEA) at 30 °C unless otherwise specified. For western blotting analysis, the *tor2-ts6* mutant was cultured at 26 °C overnight and then transferred to 30 °C for 16 h. For experiments with antifungal agents, cells were treated with 16 mM caffeine, 0.4 mM fluconazole or 0.3 μM clotrimazole in YEA medium.

### Genetic screen

Genetic screens were conducted using a heterothallic strain (*mat1M-smt0*) harbouring the *raf1[R576H]* mutation and a deletion of the local silencer (*REIIΔ*) adjacent to the *mat2P* cassette (Fig. 1a). In addition, the strain contained a *ura4+* reporter downstream of *mat2P* (*mat2P::ura4+*). Exponentially growing cells were treated with the chemical mutagen methylnitronitrosoguanidine at a concentration of 0.5 mg ml[−1] for 60 min at room temperature or irradiated with ultraviolet light (UV) at 100 J m[−2] using UV crosslinker. Colonies formed on YEA medium were replica plated onto PMG-5S medium and assayed for haploid meiosis by means of iodine staining. Light colonies were restreaked to single colonies and assessed for growth on −Ura and FOA media. Suppressor mutants were subjected to whole-genome sequencing using the Illumina NextSeq500 platform. Genomic sequencing reads were quality trimmed using fastp[61] and aligned to the *S. pombe* ASM294v2.30 reference sequence[62] with the BWA aligner[63] using default parameters. Duplicate reads in the resulting BAM files were marked using Picard tools (http://broadinstitute.github.io/picard/) 'MarkDuplicates'. Mutations were called from the duplicate-marked BAM files using samtools 'mpileup' and subsequently processed with bcftools[64] to generate a single VCF file[65] containing mutations identified in the WT and mutant genomes. Mutation impacts were predicted using SnpEff[66]. Variations that were present in the mutants but absent in the WT controls and were predicted to have 'HIGH' or 'MODERATE' impact were flagged for further investigation.

### Serial dilution assay for heterochromatic silencing

To assess the silencing of the *ura4+* reporter in *mat1M-smt0 REIIΔ mat2P::ura4+* cells, tenfold serial dilutions of cultures were spotted on uracil-deficient medium and PMG-5S medium supplemented with FOA. The derepression of *mat2P* was evaluated by exposing the cells to iodine vapour. Cells undergoing haploid meiosis on PMG-5S medium, which results from *mat2P* derepression, accumulate a starch-like compound and stain dark brown when exposed to iodine vapour. Images of serial dilution plates were presented with Adobe Photoshop v.22.4.2 and Adobe Illustrator v.2024.

### RT−qPCR

Total RNA was extracted from $2 \times 10^8$ cells using the MasterPure Yeast RNA Purification Kit (Biosearch Technologies) according to the manufacturer's instructions. Complementary DNA (cDNA) was synthesized from 1 μg of DNase-treated RNA using SuperScript III reverse transcriptase with gene-specific primers. The resulting cDNA was subjected to quantitative PCR (qPCR) using iTaq Universal SYBR Green Supermix (Bio-Rad), following the manufacturer's instructions. The *act1+* gene served as the internal control for normalization. Oligonucleotides used for qPCR are listed in Supplementary Table 3.

### ChIP−qPCR

For the H3K9me3 ChIP assay, $5 \times 10^8$ cells were crosslinked with 3% paraformaldehyde (PFA) for 30 min at room temperature, followed by quenching with 125 mM glycine. For Swi6 or Raf2 or H3K14ub ChIPs, $1 \times 10^9$ of cells were incubated for 2 h at 18 °C, fixed in 3% PFA and washed in ice-cold PBS. Cells were treated with 10 mM dimethyl adipimidate at room temperature for 45 min.

Fixed cells were resuspended in ChIP lysis buffer (50 mM HEPES/KOH pH 7.5, 140 mM NaCl, 1 mM EDTA, 1% Triton X-100, 0.1% sodium deoxycholate) supplemented with 1 mM PMSF and protease inhibitors (cOmplete Mini Protease Inhibitor Cocktail, Roche) and lysed by bead-beating (Biospec Mini-Beadbeater-16). For H3K14ub, ChIP lysis buffer was also supplemented with DUB inhibitor 100 μM PR-619 and 100 mM *N*-ethylmaleimide. The lysates were sonicated with a Diagenode Bioruptor for 14 cycles on medium power setting (30 s ON, 30 s OFF) to shear DNA into 0.4−0.6 kilobase (kb) fragments. Cellular debris was removed by centrifugation and 2% of the supernatant was kept for whole cell extract input control. The remaining lysates were precleared with protein A and protein G agarose beads for 1 h at 4 °C. After bead removal, lysates were incubated with the appropriate antibody (2−10 μg) overnight at 4 °C with slow rotation. Antibodies used include anti-H3K9me3 (Abcam), anti-c-Myc (Santa Cruz), anti-H3K14ub, as described in ref. 37, and custom affinity-purified anti-Swi6. Prewashed protein A and protein G agarose beads were added to the lysates and incubated for 4 h at 4 °C. The beads were sequentially washed, twice with wash buffer I (50 mM HEPES/KOH pH 7.5, 140 mM NaCl, 1 mM EDTA, 1% Triton X-100, 0.1% sodium deoxycholate), twice with wash buffer II (50 mM HEPES/KOH pH 7.5, 500 mM NaCl, 1 mM EDTA, 1% Triton X-100, 0.1% sodium deoxycholate), twice with wash buffer III (10 mM Tris/HCl pH 8.0, 250 mM LiCl, 0.5% IGEPAL, 0.5% sodium deoxycholate, 1 mM EDTA) and once with TE buffer (50 mM Tris/HCl pH 8.0, 10 mM EDTA). The chromatin was eluted from the beads twice with 50 μl of elution buffer (50 mM Tris/HCl pH 8.0, 10 mM EDTA, 1% SDS) at 65 °C for 30 min each, with shaking (roughly 1,200 rpm). Eluted chromatin and input control samples were decrosslinked overnight at 65 °C. Samples were treated sequentially with RNase A (10 μg, 2 h, 37 °C) and Proteinase K (20 μg, 2 h, 37 °C), and subsequently purified using the Qiagen PCR purification kit. qPCR was conducted with iTaq Universal SYBR Green Supermix (Bio-Rad) according to the manufacturer's instructions. Fold enrichment was calculated using the ΔΔCt method, with *leu1+* serving as the reference. Oligonucleotide sequences for qPCR are provided in Supplementary Table 3.

### ChIP−seq and data processing

Genome-wide analysis of ChIP DNA was conducted as previously described in ref. 9. One ng of DNA was used for library preparation using the NEBNext Ultra II DNA Library Prep Kit for Illumina (New England Biolabs) following the manufacturer's instructions. Libraries were purified with AMPure XP magnetic beads (Beckman Coulter), and their quality was analysed on the TapeStation System 4150 (Agilent) before sequencing using the Illumina MiSeq sequencing platform. Sequenced reads were quality trimmed with fastp[61] and aligned with the BWA aligner[63] to the *S. pombe* ASM294v2.30 reference sequence[62] to which we added a separate contig containing the sequence of the full 40-kb *mat* region. Both the Pombase provided *mat* region on chromosome 2 (chr. II:2109748−2138781) and the *mat1M* locus on the 40-kb *mat* contig (MAT 4,489−5,615) were masked from alignment. ChIP−seq alignments for strains lacking the *REII* element in the *mat* region were made to the standard genome as described but with the sequence of the *REII* element removed. Bedgraphs of ChIP enrichment over input were produced using the MACS2 (ref. 67) 'callpeaks' function to make broad calls

with options '-nomodel-extsize 147', followed by the MACS2 'bdgcmp' function to compute fold enrichment over the input background. As the sequence of the constitutively euchromatic *mat1* locus is identical to that of its counterpart within silent *mat3*, the euchromatic copy of *mat1* was masked when aligning reads to prevent read misalignment. The Integrative Genomics Viewer[68] was used to plot ChIP–seq data.

### RNA-seq analysis

Transcriptome analysis was performed as previously described in ref. 30. Briefly, total RNA was isolated using a MasterPure Yeast RNA Purification Kit (Lucigen) and then ribosomal RNA (rRNA) was removed using a Ribo-Zero magnetic gold rRNA removal kit (yeast; Illumina). The library was constructed using a ScriptSeq v.2 RNA-seq library preparation kit (Illumina) or NEBNext Ultra II directional RNA library prep kit for Illumina (NEB). The final library was analysed using an Agilent 2100 BioAnalyzer and sequenced on the Illumina NextSeq500 platform. Single ended short reads from RNA-seq experiments were quality trimmed using fastp[61] and aligned using the STAR aligner[69]. Variable interval bedgraphs, normalized to counts per million mapped reads and including both uniquely and multi-mapping reads, were generated by STAR and were further processed to produce 10-base pair fixed interval versions for purposes of figure construction. Read counts for transcripts were computed from BAMs using Rsubread[70] and differential expression with respect to WT controls was computed using DEGseq[71]. The Volcano plot was constructed from $\log_2$ fold-changes and $P$ values produced by DEGseq using the 'EnhancedVolcano' R library (https://github.com/kevinblighe/EnhancedVolcano).

### Northern blotting

Northern blot analysis of centromeric small interfering RNAs (siRNAs) was performed as described previously in ref. 72. Briefly, small RNAs (less than 200 nt) were purified from mid-log phase cells with mirVana microRNA isolation kit (Thermo Fisher Scientific). Then 20 μg of small RNAs were resolved on a 15% denaturing acrylamide gel and transferred to Hybond-N+ (Thermo Fisher Scientific) membrane in 0.5× TBE for 1 h at 100 V. After UV crosslinking, the membrane was hybridized with α-P$^{32}$-UTP (PerkinElmer) labelled RNA probes (roughly 50 nucleotides) corresponding to the *dg* sequence in ULTRAhyb-Oligo hybridization buffer (Thermo Fisher Scientific). The membrane was exposed and scanned using Typhoon FLA 9500 phosphor imager (GE Healthcare).

For northern blot analysis of *raf1* transcript, exponentially growing cells were resuspended in LETS buffer (100 mM LiCl, 10 mM EDTA, 10 mM Tris/HCl pH 7.5, 0.2% SDS), combined with equal volume of LETS-saturated phenol–chloroform and lysed by bead-beating. Total RNA was purified through repeated extraction with LETS-saturated phenol–chloroform and collected by precipitation after the addition of LiCl to a final concentration of 0.5 M and 2.5 volumes of 100% ethanol. The RNA was resuspended in NorthernMax formaldehyde loading dye (Invitrogen), resolved on a 1% agarose gel and transferred onto a BrightStar-Plus (Invitrogen) membrane by following the NorthernMax kit (Invitrogen) protocol. Blotted RNA was detected using α-P$^{32}$-UTP (PerkinElmer) labelled RNA probe. RNA probe was prepared by in vitro transcription with the Maxiscript T7 kit (Invitrogen). Probe was hybridized to blotted RNAs in ULTRAhyb hybridization buffer (Invitrogen) and the blots were visualized using the Typhoon FLA 9500 phosphor imager (GE Healthcare).

### RIP–qPCR

For the Myc-tagged Upf1 RIP assay, $1 \times 10^9$ cells were incubated for 2 h at 18 °C, fixed in 3% PFA and washed in ice-cold PBS. Cells were treated with 10 mM Dimethyl adipimidate at room temperature for 45 min. Fixed cells were resuspended in ChIP lysis buffer (50 mM HEPES/KOH pH 7.5, 140 mM NaCl, 1 mM EDTA, 1% Triton X-100, 0.1% sodium deoxycholate) supplemented with 1 mM PMSF, complete EDTA-free proteinase inhibitor cocktail (Roche) and 40 units of RNase inhibitor

(Thermo Fisher Scientific, AM2694) and lysed by bead-beating (Biospec Mini-Beadbeater-16). The lysates were sonicated with a Diagenode Bioruptor for 14 cycles on medium power setting (30 s ON, 30 s OFF). Cellular debris was removed by centrifugation, and 5% of the supernatant was reserved for whole cell extract input control. The remaining lysates were precleared with 0.9 mg of prewashed protein G Dynabeads (Invitrogen, 10004D) at 4 °C for 1 h. After bead removal, lysates were incubated with 5 μg of anti-c-Myc antibody (Santa Cruz, 9E10) overnight at 4 °C with slow rotation. Antibody–protein complexes were captured using 1.2 mg of protein G Dynabeads for 2 h at 4 °C. The beads were sequentially washed in 900 μl of each buffer, once with wash buffer I (50 mM HEPES/KOH pH 7.5, 140 mM NaCl, 1 mM EDTA, 1% Triton X-100, 0.1% sodium deoxycholate), once with wash buffer II (50 mM HEPES/KOH pH 7.5, 500 mM NaCl, 1 mM EDTA, 1% Triton X-100, 0.1% sodium deoxycholate), once with wash buffer III (10 mM Tris/HCl pH 8.0, 250 mM LiCl, 0.5% IGEPAL, 0.5% sodium deoxycholate, 1 mM EDTA) and once with TE buffer (50 mM Tris/HCl pH 7.0, 10 mM EDTA). Each wash was done for 7 min at slow rotation. Beads were eluted twice in 75 μl of RIP elution buffer (50 mM Tris/HCl pH 8, 10 mM EDTA, 300 mM NaCl, 1% SDS) at 37 °C for 10 min. To 50-μl input samples, 100 μl of RIP elution buffer was added to make a final volume of 150 μl. Then 20 μg of proteinase K (Thermo Fisher Scientific, AM2548) was added to both immunoprecipitation and input samples, and the mixtures were incubated at 37 °C for 1 h followed by de-crosslinking at 65 °C for 1 h. The samples were then extracted with phenol–chloroform, precipitated with ethanol and the pellet was resuspended in 80 μl of DEPC-treated water. The samples were further treated with 20 units of RNase-free DNase I (Thermo Fisher Scientific, AM2222) at 37 °C for 1 h, extracted with phenol–chloroform and ethanol precipitated as above. Immunoprecipitation and input samples were resuspended in 20 μl and 50 μl water, respectively. RNA was used for performing RT–qPCR as mentioned above. Fold enrichment was calculated using the ΔΔCt method, with *act1*+ serving as the reference. Oligonucleotide sequences for qPCR are provided in Supplementary Table 3.

### Western blotting

The $4 \times 10^7$ mid-log phase *S. pombe* cells were lysed in 20% trichloroacetic acid by bead-beating (Biospec Mini-Beadbeater-16). The precipitated proteins were washed with ethanol. For Flag-Clr4, cells were resuspended in 150 μl TBS (50 mM Tris/HCl pH 7.5, 150 mM NaCl), heated for 5 min at 95 °C and lysed by bead-beating. Lysate was resuspended in SDS–PAGE sample buffer, and resolved using polyacrylamide gel electrophoresis. Anti-c-Myc 9E10 (Santa Cruz, sc-40), anti-Flag M2 (Sigma Aldrich, F3165) and mouse IgG HRP-linked (GE Healthcare, NA931) antibodies were used for probing the epitope-tagged proteins. For endogenous proteins, affinity-purified anti-Swi6 (in-house), antibody specific for Gad8 phosphorylated at Ser546 (gift from R. Weisman) and Rabbit IgG HRP-linked (GE Healthcare, NA934) antibodies were used. Primary antibody dilution was 1:1,000 and secondary antibody dilution was 1:2,500.

### Glycerol-gradient fractionation

The $1 \times 10^9$ *S. pombe* cells grown to mid-log phase were pelleted and flash frozen in liquid nitrogen. The frozen cells were lysed using a CryoMill (Retsch) with the parameters: three cycles at 30 Hz for 1 min followed by 5 Hz for 1 min, and the resulting cell powder was resuspended in lysis buffer (20 mM Tris/HCl, pH 8.0; 2 mM EDTA; 1% IGEPAL; 2 mM β-mercaptoethanol; 137 mM NaCl and 2× complete proteinase inhibitor). The lysate was centrifuged at 27,000g at 4 °C for 30 min, and protein concentrations of the supernatant were measured using the Bradford assay. A discontinuous glycerol gradient (1.8 ml) was prepared by layering glycerol solutions with decreasing concentrations (200 μl each of 50%, 45%, 40%, 35%, 30%, 25%, 20%, 15% and 10% glycerol) in lysis buffer within a polyallomer ultracentrifuge tube (Beckman Coulter). The gradient was allowed to form at room temperature for 2 h, followed

by cooling at 4 °C for an extra 2 h. Subsequently, 120 µl of whole cell extract was loaded atop the gradient. The tubes were inserted into precooled buckets and centrifuged at 35,000 rpm in a Beckman TLS-55 rotor at 4 °C for 19 h. Fractions (150 µl) were collected starting from the top of the gradient, and 10 µl from each fraction was resolved on a 4–12% SDS–PAGE Bis-Tris gel, followed by western blotting to detect specific proteins.

## Immunoprecipitation analysis

A 2-l culture of *S. pombe* was harvested at an optical density at 600 nm ($OD_{600}$) of 1. Cells were washed with ice-cold distilled water and resuspended in 2 ml of lysis buffer (300 mM HEPES/KOH pH 7.6; 100 mM KCl; 2 mM EDTA; 2 mM PMSF; 1 mM DTT (dithiothreitol); 0.2% IGEPAL; Roche complete mini protease inhibitor cocktail) and flash frozen in liquid nitrogen as nuggets. The frozen cell nuggets were then lysed using a CryoMill (Retsch) with the following parameters: nine cycles at 30 Hz for 1 min followed by 5 Hz for 1 min. The resulting cell powder was resuspended in 10 ml of lysis buffer. The lysate was centrifuged at 4,000 rpm for 10 min, and the supernatant was further clarified by ultracentrifugation at 27,000*g* for 1 h at 4 °C. Two hundred microlitres of M2-Flag beads (Sigma Aldrich, A2220) were washed three times with wash buffer (150 mM HEPES/KOH pH 7.6, 250 mM KCl, 1 mM EDTA, 1 mM PMSF, 0.5 mM DTT, 0.1% IGEPAL) and subsequently added to the clarified lysate. After 2 h of incubation at 4 °C, beads were washed twice with wash buffer, followed by two washes with AC buffer (20 mM HEPES/KOH, pH 7.6, 200 mM KCl, 1 mM EDTA, 2 mM $MgCl_2$, 0.5 mM DTT, 0.1% IGEPAL). Beads were resuspended in 200 µl of AC buffer containing 500 µg ml$^{-1}$ Flag peptide for elution of bound protein. After overnight incubation at 4 °C, proteins in the eluate were precipitated by adding 10% trichloroacetic acid on ice for 10 min. The protein pellet was washed with acetone, air-dried, resuspended in 200 µl of SDS–PAGE sample buffer, boiled at 95 °C for 5 min and analysed by western blotting.

## Ectopic heterochromatin silencing assay

The reporter strains contain *6xtetO-ade6*$^+$ inserted at the *ura4*$^+$ locus and *Pnmt81-tetR-2xflag-clr4*$^+$ integrated at the *leu1*$^+$ locus. Strains were grown in Edinburgh Minimal Medium (EMM) liquid media for 48 h to induce TetR-Clr4 expression. Expression and heterochromatization of the *ade6*$^+$ reporter were assayed by serial dilution and ChIP–qPCR analyses, respectively. Tenfold serial dilutions were spotted on low-adenine (7.5 mg l$^{-1}$) EMM medium lacking thiamine (*tetR-clr4*$^{ON}$) or supplemented with 5 mg l$^{-1}$ thiamine (*tetR-clr4*$^{OFF}$), and *ade6*$^+$ expression status was determined by observing colony coloration. White colonies indicate *ade6*$^+$ expression, whereas *ade6*$^+$-repressed cells form red colonies. H3K9me was analysed in cells grown for 48 h in EMM and collected at $OD_{600}$ of 0.5 (establishment, *tetR-clr4*$^{ON}$ cells). For heritability of H3K9me, cells were further cultivated in liquid EMM supplemented with tetracycline (2.5 mg l$^{-1}$) and thiamine (5 mg l$^{-1}$) for 10 generations and fixed with 3% PFA at $OD_{600}$ of 0.5. Fixed cells were processed for ChIP as described earlier. Oligonucleotides used for qPCR are listed in Supplementary Table 3.

## Live-cell imaging

Cells were grown overnight at 26 °C in PMG-5S medium until they reached logarithmic phase. For Halo tagged proteins, cells were stained with 25 nM Janelia Fluor 646 HaloTag ligand (Promega) for 1 h. For *mat2P::GFP* imaging, YEA medium was used. Cells were mounted on a 2% agarose pad formed on a glass slide. Images were acquired on a DeltaVision Elite microscope (Leica) with a ×100, 1.35 numerical aperture oil lens (Olympus). Optical *z* sections were acquired (0.2-µm step size, 20 sections) for each field. Images were deconvolved and all *z*-stacks were projected into a single-plane as maximum-intensity projections. Fiji (ImageJ2 v.1.53, National Institutes of Health (NIH)) was used for processing the images.

## Image acquisition for SMT

Cells were grown in PMG-5S medium to mid-log phase and stained with 100 pM of Janelia Fluor JFX650 HaloTag ligand (JFX650 Promega) for 1 h at 30 °C. Cells were then briefly washed twice, resuspended in fresh media, mounted on a chambered coverglass (Lab Tek II) and covered by a 1% agarose patch. Imaging was performed on custom-built HiLO microscope[73]. This custom-built microscope from the CCR, LRBGE Optical Microscopy Core facility is controlled by µManager v.2.0 software (Open Imaging Inc.), equipped with an Okolab state top incubator for temperature control (30 °C) with an objective heater, a ×100, 1.49 numerical aperture objective (Olympus Scientific Solutions), 647-nm and 488-nm lasers (Coherent OBIS), and EM-CCD cameras (Evolve 512 Delta, Photometrics). Images for GFP and JFX650 (for HaloTag) were acquired simultaneously with an exposure time of 100 ms, and a time-lapse interval of 200 ms for a total of 2 min, with laser powers of 3 mW for the 647-nm laser and 100 mW for the 488-nm laser.

## SMT analysis

The custom-made software TrackRecord[74] in MATLAB (v.R2024a, The MathWorks Inc.) was used for analysis (https://sourceforge.net/projects/single-molecule-tracking/). Briefly, to analyse each time series, data were filtered using top-hat, Wiener and Gaussian filters. A region of interest was defined to encompass the nuclei based on the nuclear GFP staining, then nuclear particles were detected, fitted to two-dimensional Gaussian function for subpixel localization and finally tracked using a nearest neighbour algorithm[75]. The tracking parameters were as follows: window size for particle detection 3 pixels, maximum frame-to frame displacement of 6 pixels, shortest track 2 frames and gaps to close 2. Diffusion coefficients of individual tracks were estimated by calculating the mean-squared displacement over time and fitting the first four points in the curve to a line, the slope of which gives the average diffusion coefficient. Only those fits whose confidence intervals included only positive numbers were retained for the analysis.

Track segments were classified into distinct diffusive states using pEM (v.2)[76]. Assumptions about the type of diffusion shown by the tracked particles are not required with pEM and the number of diffusive states can be deduced from the analysis. pEM analysis requires all analysed tracks to be of the same length, and it is preferable to use shorter tracks that do not include transitions between states. Therefore, tracks were split into seven frame segments and the pEM classification analysis was performed on the set of all these track segments. The minimum number of states for the system to converge to was set at two and the maximum at seven. If the optimal number of states that the analysis converged to was seven, the algorithm was rerun with a higher number of maximum states. The number of reinitializations was set to 50 with 200 perturbation trials. The maximum number of iterations was 1,000 with a convergence criterion for the change of log likelihood of 10$^{-7}$. The number of features for the covariance matrix was set to three for tracks of length seven. A motion blur coefficient was calculated as $(1/6)(\Delta e/\Delta t)$, where $\Delta e$ corresponds to the exposure time and $\Delta t$ the acquisition interval.

Survival distributions were calculated from the immobile portions of particle tracks as in ref. 77. Immobile segments of tracks were identified on the basis of their displacements compared with HaloTag-H2B data, which was acquired with identical imaging conditions to the Raf2. Because the motion of H2B includes the thermal motion of chromatin, it can be used to determine the maximum distance a DNA-bound molecule will be allowed to move. Therefore, we used the jump distribution of H2B tracks to define three parameters: $R_{min}$, $R_{max}$ and $N_{min}$. $R_{min}$ is the maximum distance for which histone molecules can move between two consecutive frames, calculated as the 99th percentile for H2B frame-to-frame displacements. $R_{max}$ is the maximum distance for which bound histone molecule may move in $N_{min}$ frames, calculated as

the 99th percentile of displacements of histone molecules after $N_{min}$ frames. $N_{min}$ is the minimum frame number of a shortest bound track, and is calculated to obtain a low probability, $P = 0.01$, that a diffusing molecule is classified as bound using the following equation:

$$P(R_{min}, N_{min}) = \left(1 - e^{-\frac{R_{min}^2}{4D\Delta t}}\right) N_{min}$$

The photobleaching rate was estimated as the slowest exponential decay obtained from fitting the H2B survival distribution to a triple exponential decay. The survival distribution of the Raf2 was corrected for photobleaching as follows ($S(t) = SE(t)/\gamma(t)$, where $S(t)$ corresponds to the survival distribution after photobleaching correction, $SE(t)$ the empirical survival distribution and $\gamma(t)$ is the normalized photobleaching decay).

## In vitro affinity pulldown

Recombinant His6-Clr4, glutathione S-transferase (GST) and GST-Raf1 (amino acids (aa) 211–638) were expressed and purified from *Escherichia coli* (BL21). His6-Clr4 was purified by Ni-bead affinity pulldown followed by high-performance liquid chromatography size fractionation (GenScript). GST and GST-Raf1 (aa 211–638) were purified by affinity pulldown using Glutathione Sepharose 4B resin (Cytiva, 17075601). Purified His6-Clr4 was added to GST or GST-Raf1 (aa 211–638) bound Glutathione Sepharose beads in affinity binding buffer (20 mM Tris/HCl pH 7.5, 100 mM NaCl, 1 mM EDTA, 0.05% NP-40, 2 mM DTT and complete protease inhibitors) at a protein ratio of 4:1. After 60 min of incubation with gentle rotation at 4 °C, beads were pulled down by a short spin at 500*g*. Beads were washed twice with 500 µl of affinity binding buffer and resuspended in SDS sample buffer. Bound proteins were analysed by SDS–PAGE and western blotting for Clr4 with an affinity-purified rabbit α-Clr4 antibody (dilution 1:1,000) and Alexa-Fluor 647 labelled secondary antibody (Invitrogen, A21246) (dilution 1:2,500). Fluorescence signals were captured by scanning using Typhoon FLA 9500 (GE Healthcare).

## Quantification and statistical analysis

Quantification and statistical tests used are described in the figure legends. The *n* represents the number of independent biological replicates. GraphPad Prism v.10 software was used to plot all the graphs and calculate statistical significance.

## Materials availability

Strains and plasmids generated in this study are available from the corresponding author upon reasonable request.

## Reporting summary

Further information on research design is available in the Nature Portfolio Reporting Summary linked to this article.

## Data availability

ChIP–seq and RNA-seq datasets are deposited in the Gene Expression Omnibus with accession numbers GSE280646 and GSE280607. Source data are provided with this paper.

## Code availability

The software used to analyse the data is listed in the Methods and is publicly available. Picard tools is available through GitHub at http://broad-institute.github.io/picard/; the EnhancedVolcano R library is available through GitHub at https://github.com/kevinblighe/EnhancedVolcano and the software TrackRecord[74] in MATLAB is available through sourceforge at https://sourceforge.net/projects/single-molecule-tracking/.

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

**Acknowledgements** We are grateful to R. Weisman for the generous gift of antibody against Gad8 phosphorylated at Ser546; K. Zhang, R. Pandian, A. Anil, R. Sahu and L. Sun for helpful contributions; members of the Grewal laboratory for valuable discussions and J. Barrowman for editing the paper. This study used the computational resources of the NIH-HPC. This research was supported by the Intramural Research Program of the NIH. The contributions of the NIH authors are considered Works of the United States Government. The findings and conclusions presented in this paper are those of the authors and do not necessarily reflect the views of the NIH or the US Department of Health and Human Services.

**Author contributions** S.I.S.G., Y.W. and B.B. conceived the project and designed experiments. B.B., Y.W., A.K.P., M.Z., J.D., S.J., H.X., D.V. and H.D.F. performed experiments. D.A.B. and T.S.K. helped with SMT data acquisition and data analyses, D.W. performed bioinformatic analyses. H.Q. and J.W. generated anti-H3K14ub antibody. All authors contributed to data interpretation. B.B. and S.I.S.G. wrote the paper with input from all authors. All authors read and approved the paper.

**Competing interests** The authors declare no competing interests.

**Additional information**
**Correspondence and requests for materials** should be addressed to Shiv I. S. Grewal.

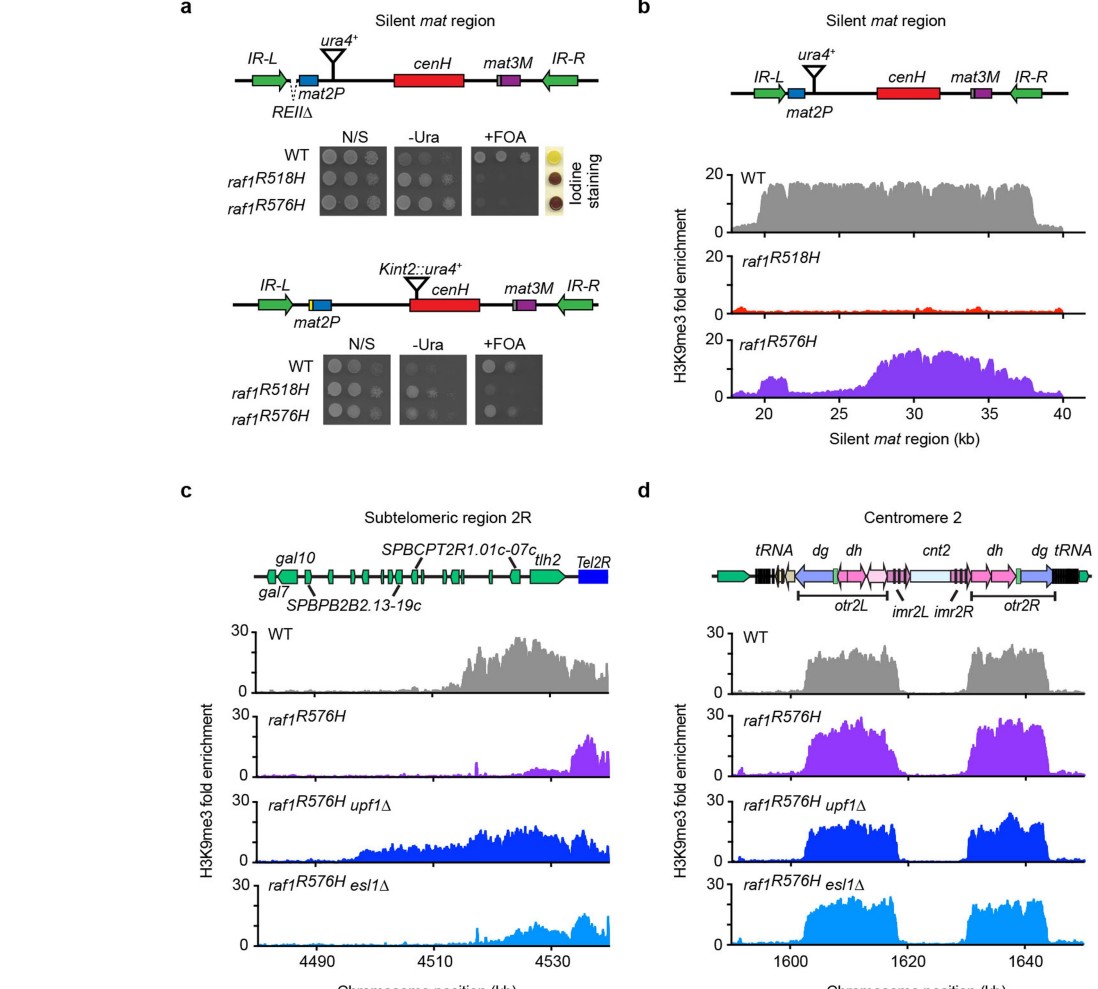

**Extended Data Fig. 1 | Mutations in components of the NMD pathway affect heterochromatin propagation. a**, Heterochromatic silencing in *raf1^R518H* or *raf1^R576H* mutant cells. Serial dilutions on indicated media were used to assess *mat2P::ura4^+* (top) or *Kint2::ura4^+* expression (bottom). Haploid meiosis was assessed by iodine staining. **b, c, d**, ChIP-seq analysis of H3K9me3 distribution across the silent *mat* region (b), the subtelomeric region 2 R (c) and centromere 2 (d) in the indicated strains. Data are representative of two independent experiments. Source data are provided.

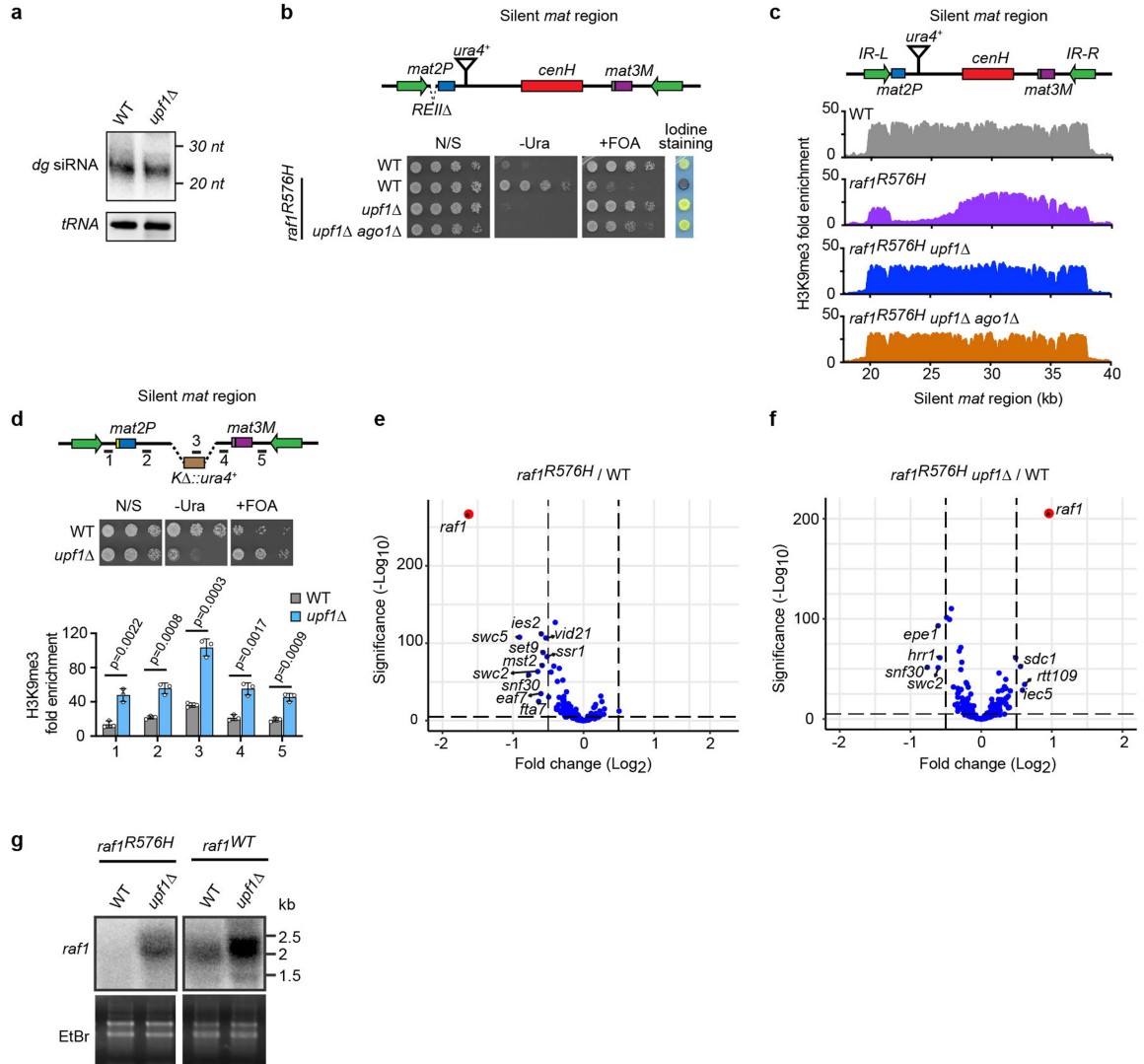

**Extended Data Fig. 2 | NMD affects heterochromatin independently of the RNAi pathway. a**, Northern blot analysis of *dg* siRNA in WT and *upf1Δ* cells. *tRNA* was used as a loading control. **b**, Heterochromatic silencing in *raf1^R576H* mutant cells with simultaneous loss of NMD and RNAi components. Serial dilution evaluated *mat2P::ura4^+* expression, and iodine staining assessed haploid meiosis. **c**, ChIP-seq analysis of H3K9me3 distribution across the silent *mat* region in the indicated strains. **d**, Effect of *upf1Δ* on *KΔ::ura4^+* expression. Serial dilutions were plated on indicated media to assess *KΔ::ura4^+* expression (top). ChIP-qPCR analysis of H3K9me3 across the *mat* region (bottom) (data are presented as mean ± SD; n = 3 independent experiments, *P* value by two-stage step-up Benjamini, Krieger, and Yekutieli unpaired t test). **e, f**, Volcano plot from RNA-seq analyses showing the fold change in the expression of heterochromatin-related genes in the *raf1^R576H* over WT cells (e), and *raf1^R576H upf1Δ* double mutant over WT cells (f) (Log_2FC and two-sided *P* values were computed using the MARS model as implemented in DEGseq. The *P* value significance threshold is set to 1e-5 (-log10 (*P* value) = 5) to correct for multiple comparisons). **g**, Northern blot analysis of *raf1* transcript in the indicated strains. Ribosomal RNA served as the loading control. Data are representative of two independent experiments. Source data are provided.

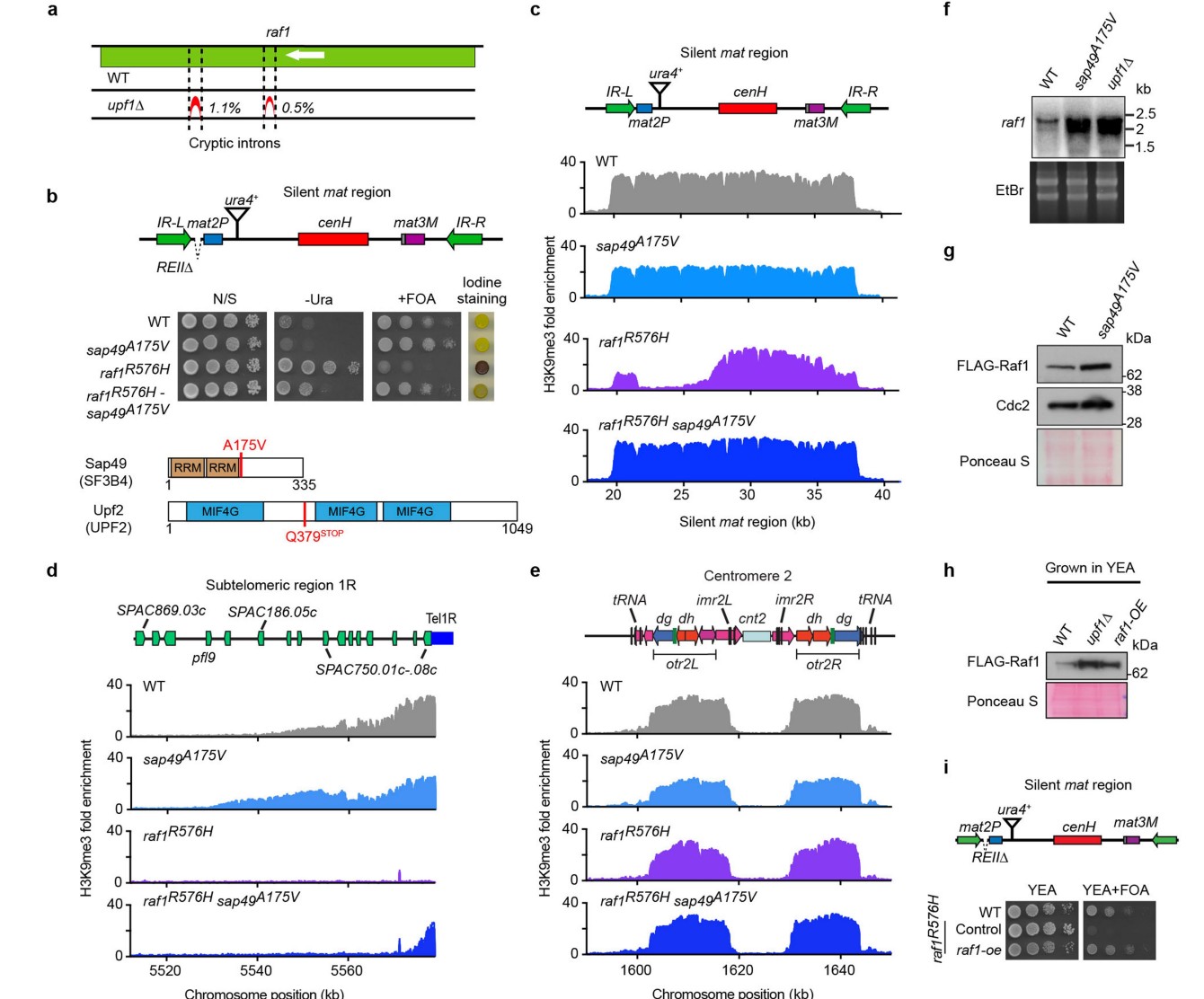

**Extended Data Fig. 3 | Splicing factor mutation mimics NMD loss in causing *raf1* upregulation. a**, RNA-seq splice junctions reveal cryptic introns in *raf1*. Numbers indicate the percentage of spliced reads. **b**, Mutation in the Sap49 splicing factor or the NMD component Upf2 suppresses the heterochromatic silencing defect in *raf1^R576H* cells. Serial dilutions assessed *mat2P::ura4^+* expression, and iodine staining assessed haploid meiosis (top). Domain organizations of Sap49 and Upf2 are shown (bottom). **c, d, e**, ChIP-seq profiles of H3K9me3 across the silent *mat* region (c), the subtelomeric region 1R (d), and centromere

2 (e) in the indicated strains. **f**, Northern blot of *raf1* transcripts in the indicated strains; ribosomal RNA served as the loading control. **g, h**, Western blot analysis of Raf1 in the indicated strains. Cdc2 and Ponceau S served as loading controls. **i**, Expressing Raf1 in *raf1-oe* (*Pnmt1-raf1*) cells at levels comparable to those in *upf1Δ* cells is sufficient to suppress the heterochromatic silencing defect in *raf1^R576H* cells. Serial dilutions on the indicated media were used to assess *mat2P::ura4^+* expression. Data are representative of two independent experiments. Source data are provided.

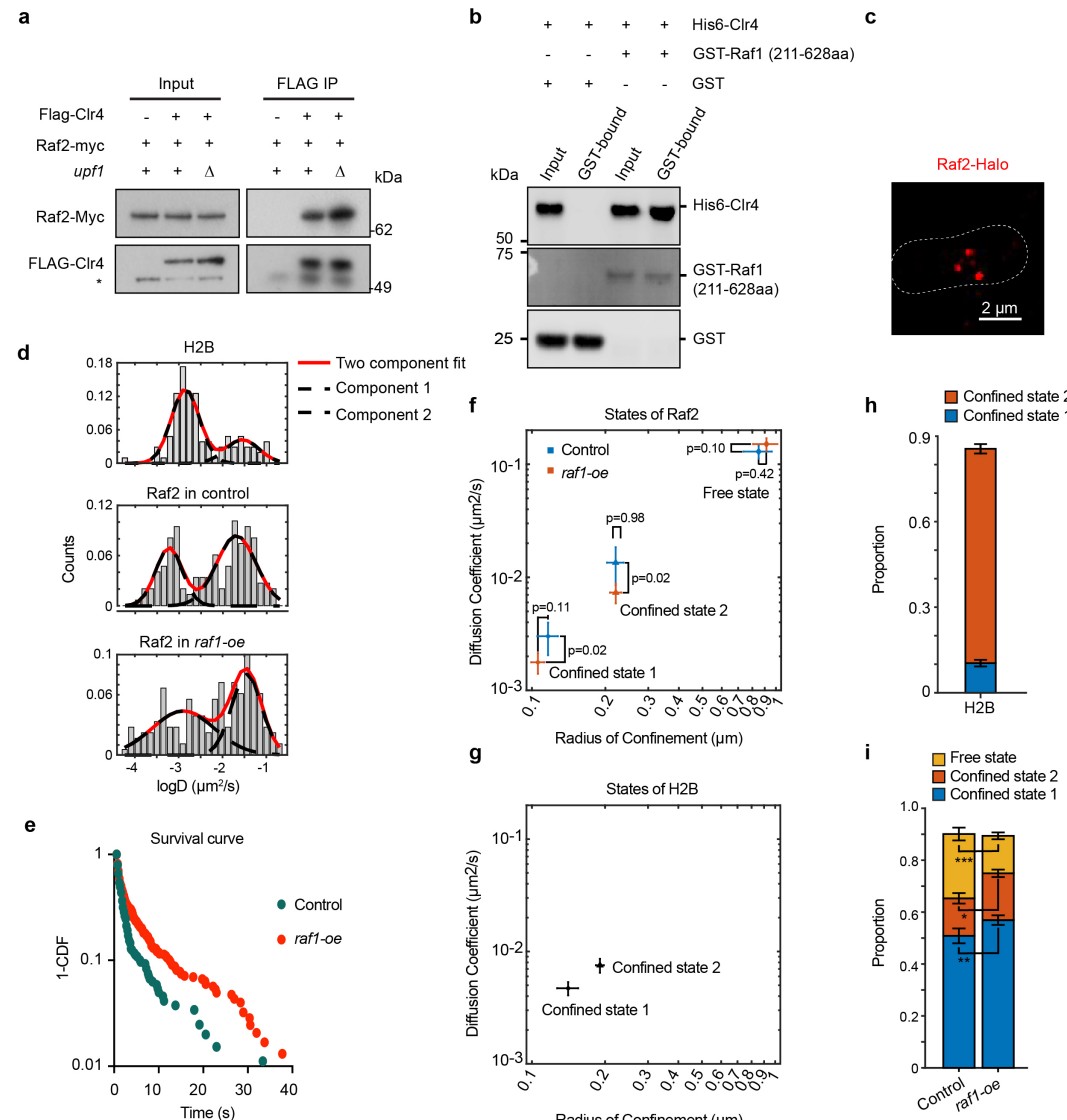

**Extended Data Fig. 4 | Raf1 overexpression enhances the association of ClrC with chromatin. a**, Co-IP of Raf2 with Clr4 upon deletion of *upf1*. * denotes the non-specific band. **b**, *In-vitro* (GST) pulldown of purified Clr4 with purified Raf1 (211-628aa). **c**, Live-cell fluorescence imaging of Raf2-Halo with Halo ligand JFX650. Scale bars, 2 μm. **d**, Distribution of diffusion coefficients (D, log$_{10}$) from individual tracks (gray bars) of H2B in WT (top), Raf2 in control (*Pnmt1-GFP*) (middle) and Raf2 in *raf1-oe* (*Pnmt1-raf1*) (bottom), all labeled with HaloTag and JFX650, and fit with a mixture of two Gaussians population (red line). Dashed black lines show the two individual sub-populations. **e**, Survival probability curves (1-CDF [cumulative distribution function]) from apparent dwell times of single-molecule chromatin-binding events of Raf2-Halo in control and *raf1-oe* cells. **f, g**, Comparison of diffusion coefficients and confinement radii for the three diffusive states of Raf2 (f) or two diffusive states of H2B (g) as obtained by pEM analysis of SMT data (*P* values from two-sided Z-test comparing). **h**, Proportion of H2B molecular states in WT (mean ± 95% CI). **i**, Proportion of Raf2 molecular states in control and *raf1-oe* cells (mean ± 95% CI, *p = 4.7 × 10$^{-3}$, **p = 5.5 × 10$^{-4}$, and ***p = 2.0 × 10$^{-12}$, *P* values from two-sided Z-test). **a, b, c**, Data are representative of two independent experiments. **d, f, i**, Data are representative of 5 independent experiments which included 310 tracks for control and 1,019 tracks *raf1-oe* cells. **d, g, h**, Data are representative of 3 independent experiments which included 547 tracks of H2B. Source data are provided.

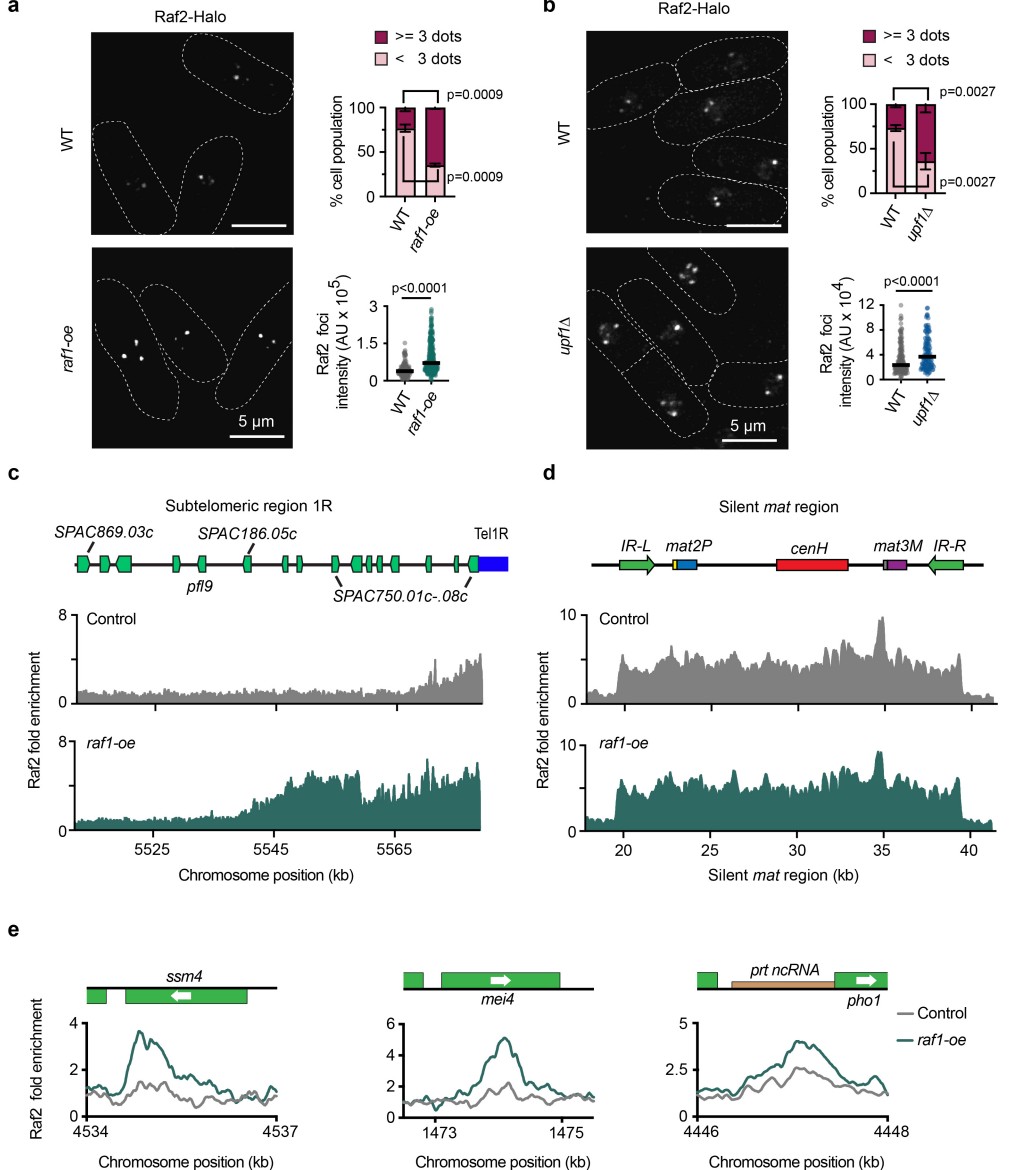

**Extended Data Fig. 5 | Increase in ClrC localization to heterochromatin regions upon Raf1 overexpression. a, b,** Fluorescence live-cell imaging of Raf2-HaloTag (JFX650) in the indicated cells and quantification of the number of Raf2 foci and their intensity. Number of foci are represented with mean ± SD, n = 3 independent experiments, *P* value by two-stage step-up Benjamini, Krieger, and Yekutieli unpaired t test (a,b, top right). Intensity of individual foci represented with median; >100 foci are represented; n = 3 independent experiments, *P* value by two-tailed unpaired t test (a,b, bottom right). **c, d, e,** ChIP-seq analysis of myc-tagged Raf2 distribution across the subtelomeric region 1 R (c), the silent *mat* region (d) and the heterochromatin islands (e) upon *raf1-oe* (*Pnmt1-raf1*). Data are representative of two independent experiments. Source data are provided.

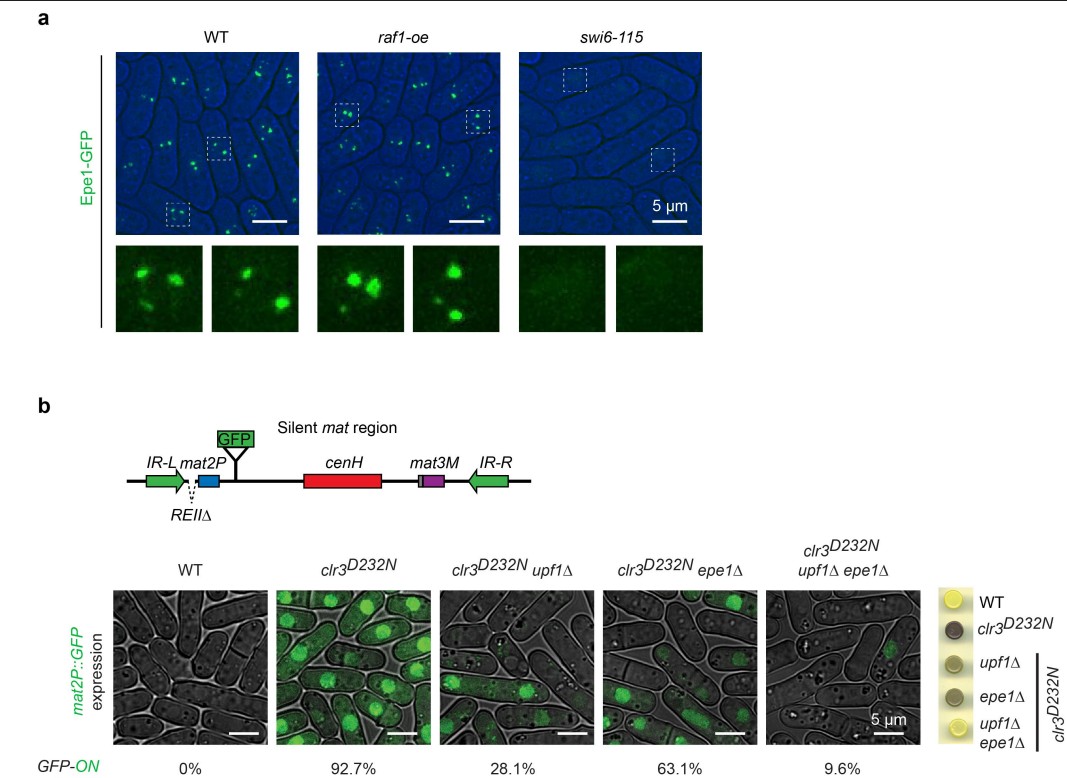

**Extended Data Fig. 6 | Raf1 regulates heterochromatin independent of Epe1. a**, Live-cell fluorescence imaging of Epe1-GFP in WT, *raf1-oe* (*Pnmt1-raf1*) and *swi6-115* mutant cells. Note that the number and intensity of Epe1-GFP foci are similar in WT and *raf1-oe* cells. Epe1-GFP foci are abolished in *swi6-115* mutant cells included as a control. Scale bars, 5 μm. Dashed boxes indicate zoomed-in areas in the bottom panel. **b**, Fluorescence imaging analysis of *mat2P::GFP* expression in the indicated strains. The percentage of cells expressing GFP (*GFP-ON*) is indicated. Scale bar, 5 μm. Haploid meiosis was assessed by iodine staining. Data are representative of two independent experiments.

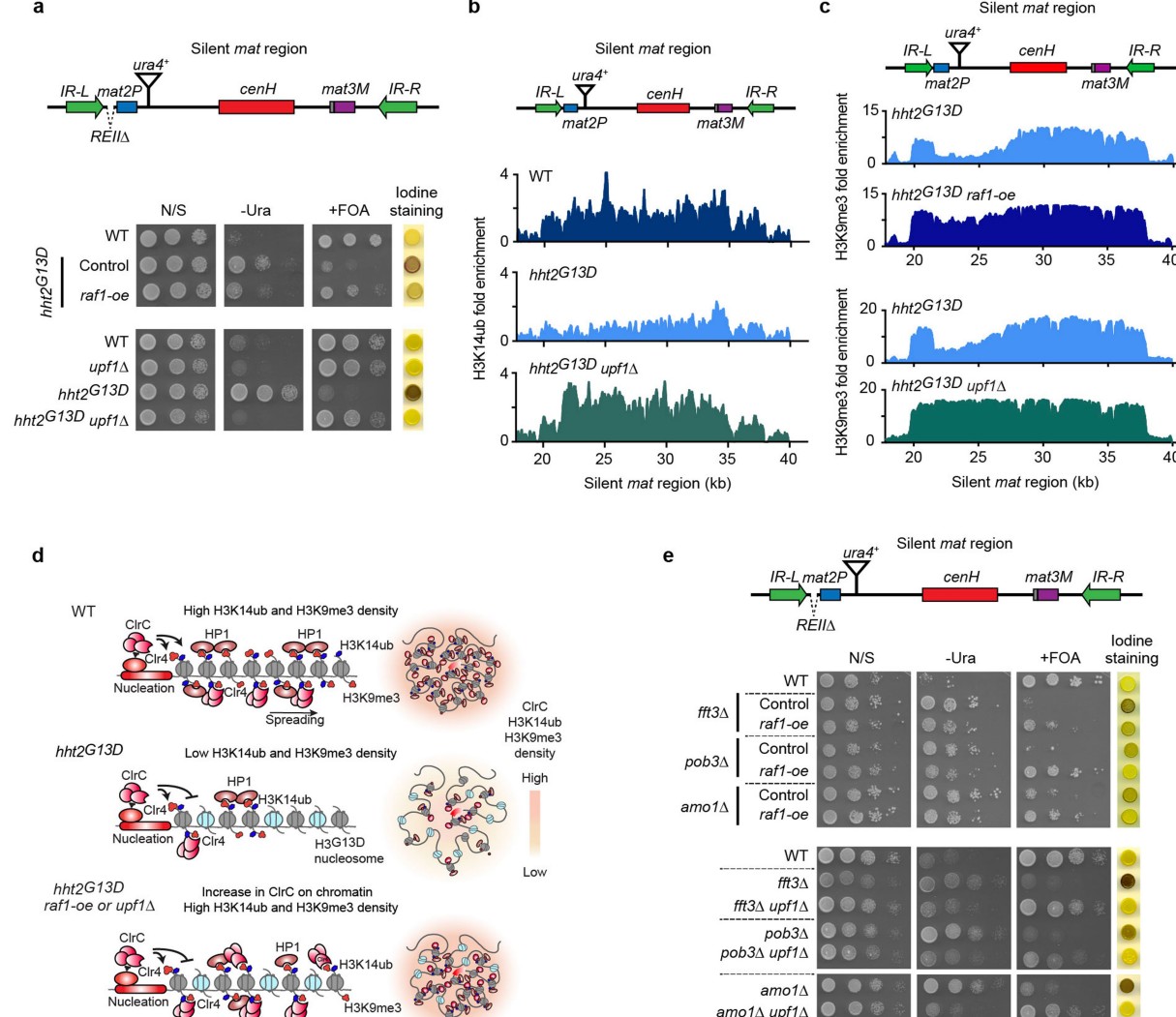

**Extended Data Fig. 7 | Increased Raf1 abundance alleviates heterochromatin propagation defects in various mutants. a**, Heterochromatic silencing upon Raf1 overexpression (*raf1-oe*; *Pnmt1-raf1*) or loss of *upf1* in *hht2^G13D^* (H3.2^G13D^) mutant cells. Serial dilution assessed *mat2P::ura4^+^* expression, and iodine staining assessed haploid meiosis. **b, c**, ChIP-seq analysis of H3K14ub (b) and H3K9me3 (c) distribution across the silent *mat* region in the indicated strains. Note that the WT H3K14ub data is the same as in Fig. 4b as the experiments were performed at the same time. **d**, Schematic illustrating restoration of heterochromatin propagation upon Raf1 overexpression or loss of Upf1 in *hht2^G13D^* cells. Heterochromatin initiated at the nucleation site through recruitment of ClrC spreads across an extended domain through the

Clr4^SUV39H^ read-write activity, requiring a critical density of H3K9me3 as well as H3K14ub (top). Incorporation of *hht2^G13D^* causes reduction not only in H3K9me3 density, but also H3K14ub, thus disrupting heterochromatin propagation (middle). Elevated Raf1 levels enhance chromatin association of ClrC, H3K14ub and H3K9me3 density, reestablishing heterochromatin spreading (bottom). **e**, Elevating Raf1 expression in *raf1-oe* or *upf1Δ* cells bypass the requirement of other factors for heterochromatin propagation. Serial dilution measured *mat2P::ura4^+^* expression, and iodine staining assessed haploid meiosis. Data are representative of two independent experiments. Source data are provided.

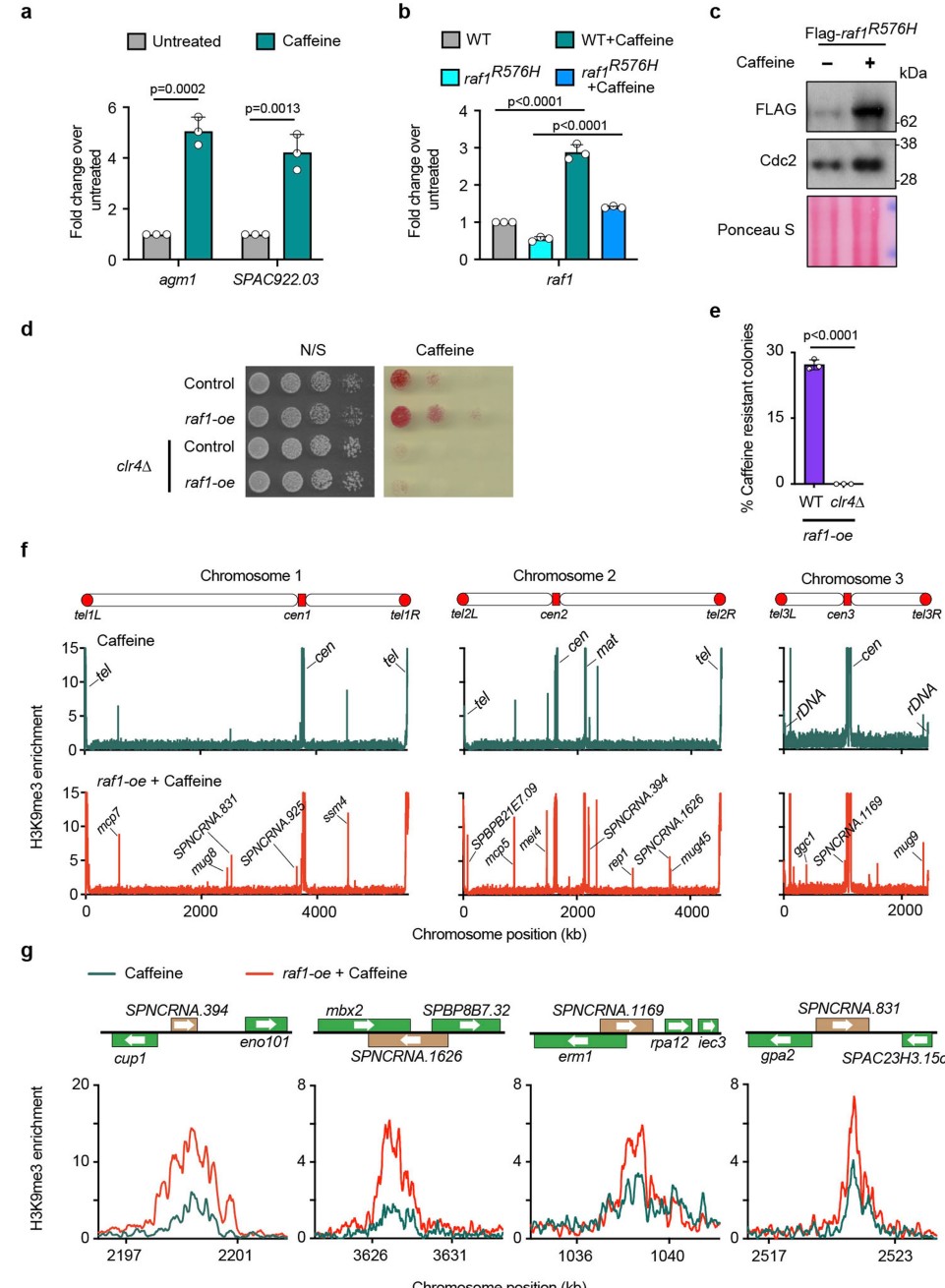

**Extended Data Fig. 8 | Increase in Raf1 abundance promotes caffeine resistance through heterochromatin formation. a, b**, Real-time qPCR analysis of indicated genes in cells treated with 16 mM caffeine for 2 weeks in WT or $raf1^{R576H}$ cells. Data are presented as mean ± SD; n = 3 independent experiments, $P$ value by two-stage step-up Benjamini, Krieger, and Yekutieli unpaired t test for (a) and one-way ANOVA followed by Tukey's multiple comparisons test for (b). **c**, Western blot analysis of Raf1 in $raf1^{R576H}$ cells treated with 16 mM caffeine for 2 weeks. Cdc2 and Ponceau S staining included as loading controls. **d**, Serial dilutions of the indicated strains were plated on nonselective (N/S) or 16 mM caffeine-containing medium. **e**, Quantification of caffeine-resistant colonies in

$raf1-oe$ cells ($Padh1-raf1$) with or without $clr4$ (data are presented as mean ± SD; n = 3 independent experiments, $P$ value by two-tailed unpaired t test). **f**, ChIP-seq analyses of H3K9me3 enrichment in cells treated with 16 mM caffeine for 2 weeks, with or without $raf1-oe$. Facultative heterochromatin islands with increased H3K9me3 levels in $raf1-oe$ cells are indicated. **g**, Representative plots of H3K9me3 distribution at facultative islands responsive to caffeine. Open reading frame (ORF) positions are indicated. Data are representative of two independent experiments. Source data are provided.

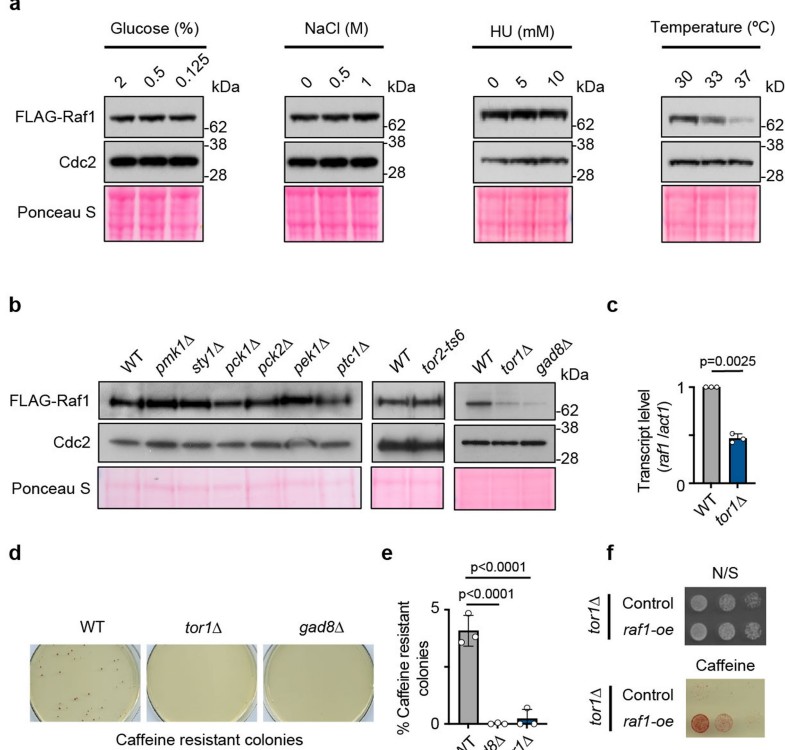

**Extended Data Fig. 9 | Tor1-Gad8 signaling axis modulates Raf1 levels and caffeine resistance. a, b**, Western blot analysis of Raf1 under stress conditions for 16 h (a) or in the indicated strains (b). **c**, Real-time qPCR analysis of *raf1* expression in WT or *tor1Δ* cells (data are presented as mean ± SD; n = 3 independent experiments, *P* value by two-tailed unpaired t test). **d**, WT, *tor1Δ*, and *gad8Δ* cells plated on 16 mM caffeine-containing medium.

**e**, Quantification of caffeine resistant colonies (data are presented as mean ± SD; n = 3 independent experiments, *P* value by one-way ANOVA followed by Dunnett's multiple comparisons test). **f**, Caffeine resistance in *tor1Δ* cells upon *raf1-oe* (*Padh1-raf1*). Serial dilutions were plated on nonselective (N/S) and 16 mM caffeine containing media. Data are representative of two independent experiments. Source data are provided.

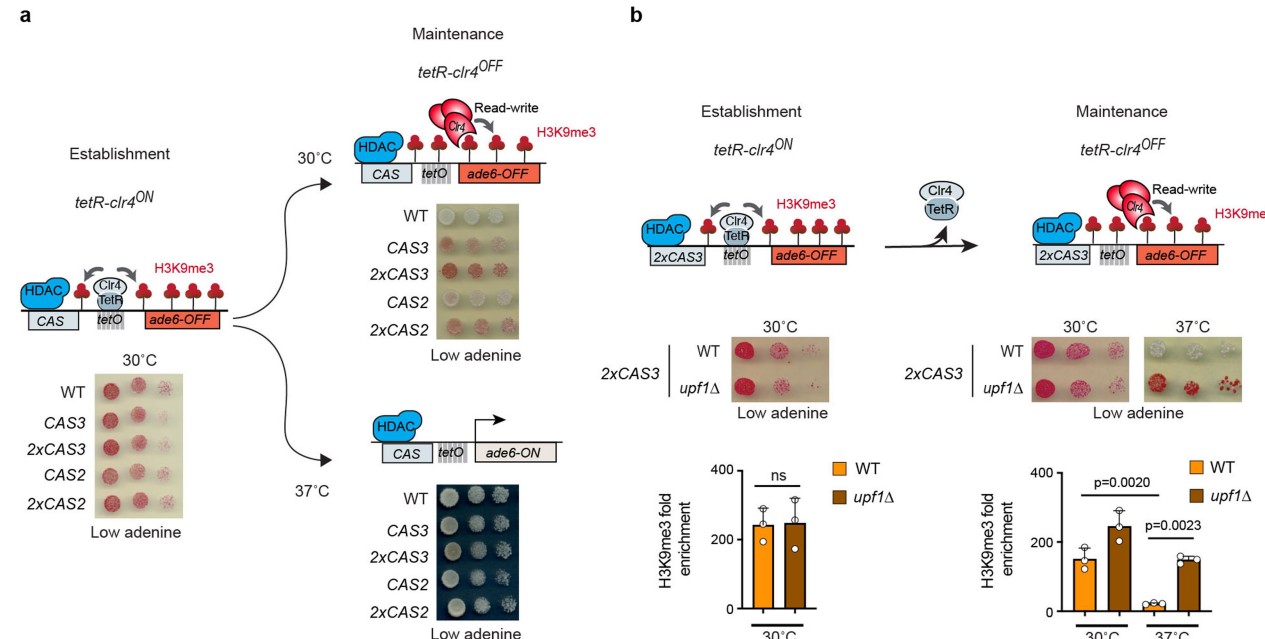

**Extended Data Fig. 10 | Elevated Raf1 expression enables epigenetic inheritance of heterochromatin under heat stress. a**, Effect of high temperature (37 °C) on heterochromatin propagation at an ectopic site in the indicated strains. Establishment of silenced heterochromatin was assessed by serial dilution in low-adenine-containing medium. TetR-Clr4 was expressed (*tetR-clr4^ON^*) from a thiamine-repressible promoter in cells containing the *tetO-ade6^+^* reporter, with or without *CAS* sites as indicated. Maintenance of *ade6^+^* gene silencing was assessed by serial dilution in thiamine-supplemented low-adenine medium (*tetR-clr4^OFF^*). **b**, Maintenance of *ade6^+^* gene silencing in cells containing *2xCAS3-tetO-ade6^+^* with or without *upf1* at 30 °C and 37 °C was assessed by serial dilution in thiamine-supplemented low-adenine medium (*tetR-clr4^OFF^*). H3K9me3 enrichment at the *ade6^+^* was determined by ChIP-qPCR (data are presented as mean ± SD; n = 3 independent experiment, *P* value by one-way ANOVA followed by Tukey's multiple comparisons test). Data are representative of three independent experiments. Source data are provided.

# Reporting Summary

## Statistics

For all statistical analyses, confirm that the following items are present in the figure legend, table legend, main text, or Methods section.

| n/a | Confirmed | |
|---|---|---|
| ☐ | ☒ | The exact sample size (*n*) for each experimental group/condition, given as a discrete number and unit of measurement |
| ☐ | ☒ | A statement on whether measurements were taken from distinct samples or whether the same sample was measured repeatedly |
| ☐ | ☒ | The statistical test(s) used AND whether they are one- or two-sided *Only common tests should be described solely by name; describe more complex techniques in the Methods section.* |
| ☒ | ☐ | A description of all covariates tested |
| ☐ | ☒ | A description of any assumptions or corrections, such as tests of normality and adjustment for multiple comparisons |
| ☐ | ☒ | A full description of the statistical parameters including central tendency (e.g. means) or other basic estimates (e.g. regression coefficient) AND variation (e.g. standard deviation) or associated estimates of uncertainty (e.g. confidence intervals) |
| ☐ | ☒ | For null hypothesis testing, the test statistic (e.g. *F*, *t*, *r*) with confidence intervals, effect sizes, degrees of freedom and *P* value noted *Give P values as exact values whenever suitable.* |
| ☒ | ☐ | For Bayesian analysis, information on the choice of priors and Markov chain Monte Carlo settings |
| ☒ | ☐ | For hierarchical and complex designs, identification of the appropriate level for tests and full reporting of outcomes |
| ☒ | ☐ | Estimates of effect sizes (e.g. Cohen's *d*, Pearson's *r*), indicating how they were calculated |

*Our web collection on statistics for biologists contains articles on many of the points above.*

## Software and code

Policy information about availability of computer code

Data collection | Microscopy imaging was done with Delta Vision Elite microscope (Leica). Single molecule tracking was performed on custom-built HiLO microscope (Mehta et al., Mol Cell, 2018). ChIP-seq data were collected using next generation Illumina sequencer Miseq. RNA-seq data were collected using next generation Illumina sequencer NextSeq 500/550. Western blot signals were acquired using the Supersignal West Pico PLUS Chemiluminescent substrate (Thermofisher) and Amersham Hyperfilm ECL developed with Konica SRX-101A. Real time PCR was performed on QuantStudio3 (applied biosystems) using iTaq Universal SYBR Green Supermix (BioRad). Northern blot was scanned using Typhoon FLA 9500 phosphor imager (GE Healthcare).

| Data analysis | Microscopy image processing, including deconvolution and maximum intensity projection, was performed using ImageJ2 1.53v. Acquisition of single molecule tracks were performed using µManager v2.0 software and were analysed by the custom-made software "TrackRecord" in MATLAB (Mazza et al., Methods Mol Biol, 2013) and is available at https://sourceforge.net/projects/single-molecule-tracking/. Segments of single molecule tracks were classified into distinct diffusive states using perturbation-Expectation Maximization (Koo et al., Phys Rev, 2016). ChIP-seq and RNA seq data were visualized in Integrated Genome Browser (IGV_2.17.4). ChIP seq reads were trimmed with fastp v1.0.1 and aligned with BWA aligner v0.7.17 software to S. pombe V2 reference sequence which is available at https://www.pombase.org/. MACS2 v2.2.7.1 callpeaks function was used to produces ChIP enrichment Bedgraphs followed by bdgcmp function to compute fold enrichment over input. Integrative Genomics Viewer v2.17.4 was used to plot enrichments along the fission yeast chromosomes. RNA seq reads were trimmed with fastp v1.0.1 and aligned with STAR aligner v2.7.11b software to S. pombe V2 reference sequence which is available at https://www.pombase.org/. Serial dilution plates were scanned using Epson Perfection V700 Photo and presented with Adobe Photoshop 22.4.2 and Illustrator 2024. The statistical tests and statistic parameters including the mean and standard deviation (SD) were calculated and plotted with Prism 10 for Mac (GraphPad Software). |
|---|---|

For manuscripts utilizing custom algorithms or software that are central to the research but not yet described in published literature, software must be made available to editors and reviewers. We strongly encourage code deposition in a community repository (e.g. GitHub). See the Nature Portfolio guidelines for submitting code & software for further information.

## Data

Policy information about availability of data

All manuscripts must include a data availability statement. This statement should provide the following information, where applicable:
- Accession codes, unique identifiers, or web links for publicly available datasets
- A description of any restrictions on data availability
- For clinical datasets or third party data, please ensure that the statement adheres to our policy

The ChIP-seq and RNA-seq data reported in this paper are available at Gene Expression Omnibus (https://www.ncbi.nlm.nih.gov/geo/) with accession number GSE280646 and GSE280607 respectively.
Source data are provided with this paper in the Source Data file.
Raw images of WB, NB and gels are provided in Supplementary Figure 1 with this paper.
S. pombe ASM294v2.30 reference sequence is available at https://www.pombase.org/.

## Research involving human participants, their data, or biological material

Policy information about studies with human participants or human data. See also policy information about sex, gender (identity/presentation), and sexual orientation and race, ethnicity and racism.

| Reporting on sex and gender | N/A |
|---|---|
| Reporting on race, ethnicity, or other socially relevant groupings | N/A |
| Population characteristics | N/A |
| Recruitment | N/A |
| Ethics oversight | N/A |

Note that full information on the approval of the study protocol must also be provided in the manuscript.

## Field-specific reporting

Please select the one below that is the best fit for your research. If you are not sure, read the appropriate sections before making your selection.

☒ Life sciences ☐ Behavioural & social sciences ☐ Ecological, evolutionary & environmental sciences

For a reference copy of the document with all sections, see nature.com/documents/nr-reporting-summary-flat.pdf

## Life sciences study design

All studies must disclose on these points even when the disclosure is negative.

| Sample size | Sample size selection was based on experience from prior studies: [Sahu et al., 2024, Mol Cell (PMID: 39096900); Wei et al., 2021, Nat Cell Biol (PMID: 33574613); Holla et al., 2020, Cell (PMID: 31883795); Mazza et al., 2013, Methods Mol Biol (PMID: 23980004) |
|---|---|
| Data exclusions | No data were excluded from analyses. |

| | |
|---|---|
| Replication | For all experiments, at least two biological replicates were performed. All attempts at replication were successful. The number of replicates is indicated in the figure legends. |
| Randomization | Randomization was not relevant to the assays conducted in this study. All cultures were grown under strictly controlled conditions and collected without any subjective selection. |
| Blinding | Investigators were not blinded during this study; experimental conditions were strictly controlled, and proper controls utilized. No subjective group allocation was required. |

# Reporting for specific materials, systems and methods

We require information from authors about some types of materials, experimental systems and methods used in many studies. Here, indicate whether each material, system or method listed is relevant to your study. If you are not sure if a list item applies to your research, read the appropriate section before selecting a response.

## Materials & experimental systems

| n/a | Involved in the study |
|---|---|
| ☐ | ☒ Antibodies |
| ☒ | ☐ Eukaryotic cell lines |
| ☒ | ☐ Palaeontology and archaeology |
| ☒ | ☐ Animals and other organisms |
| ☒ | ☐ Clinical data |
| ☒ | ☐ Dual use research of concern |
| ☒ | ☐ Plants |

## Methods

| n/a | Involved in the study |
|---|---|
| ☐ | ☒ ChIP-seq |
| ☒ | ☐ Flow cytometry |
| ☒ | ☐ MRI-based neuroimaging |

## Antibodies

| | |
|---|---|
| Antibodies used | ChIP experiments were performed by combining 1 ml of pre-cleared cell lysate with appropriate antibody: 2μg of anti-H3K9me3 (Abcam, Catalog#ab8898), 5μg of anti-c-Myc 9E10 (Santa Cruz, Catalog#sc-40), 1:100 dilution of anti-H3K14ub (from Jiemin Wong, reported in PMID: 41094145) or 5μg of rabbit anti-Swi6 (in-house affinity purified antibody, reported in PMID: 10847685, PMID: 11283354, PMID: 11498594, PMID: 15976807). Immunoprecipitation experiments were performed with M2-FLAG beads (Sigma Aldrich, Catalog#A2220). Western blot experiments were performed with anti-c-Myc 9E10 (Santa Cruz, Catalog#sc-40), anti-FLAG M2 (Sigma Aldrich, Catalog#F3165), anti-Cdc2 (Santa cruz, Catalog#sc-53217), anti-Gad8 phosphorylated at Ser546 (gift from Ronit Weisman, reported in PMID: 24344203), rabbit anti-Swi6 (in-house affinity purified, reported in PMID: 10847685, PMID: 11283354, PMID: 11498594, PMID: 15976807) or rabbit anti-Clr4 (in-house affinity purified antibody) antibody diluted at 1:1000 dilution. We used 1:2500 dilution of secondary anti-mouse IgG1 (GE Healthcare, Catalog#NA931), rabbit IgG (GE Healthcare, Catalog#NA934) HRP linked antibody or alexa-Fluor 647 labelled secondary antibody (Invitrogen, Catalog#A21246). |
| Validation | All antibodies used in this study are well-recognized clones. Antibodies have been validated both by the manufacturer and citations provided for the scientific literature. Validation data for each antibody can be accessed on the respective manufacturer's homepage. Anti-Swi6 is affinity purified in-house antibody which is reported in PMID: 10847685, PMID: 11283354, PMID: 11498594, PMID: 15976807. Anti-H3K14ub antibody is reported in PMID: 41094145. Antibodies against Gad8 phosphorylated at Ser546 is reported in PMID: 24344203. |

## Plants

| | |
|---|---|
| Seed stocks | N/A |
| Novel plant genotypes | N/A |
| Authentication | N/A |

## ChIP-seq

### Data deposition

☒ Confirm that both raw and final processed data have been deposited in a public database such as GEO.

☒ Confirm that you have deposited or provided access to graph files (e.g. BED files) for the called peaks.

| Data access links | The ChIP-seq and RNA-seq data reported in this paper are available at Gene Expression Omnibus (https://www.ncbi.nlm.nih.gov/geo/) with accession number GSE280646 and GSE280607 respectively. |
|---|---|

*May remain private before publication.*

| Files in database submission | Please see above. |
|---|---|

| Genome browser session (e.g. UCSC) | N/A |
|---|---|

## Methodology

| Replicates | Three or more biological replicates were performed for H3K9me3 ChIP-seq in wild type and raf1R576H mutant. Two biological replicates were performed for H3K9me3 ChIP-seq in raf1R518H mutant, raf1R576H upf1Δ double mutant, GFP overexpressing cells, Raf1 overexpressing cells, h3.2G13D mutant, clr3Δ mutant, caffeine treated cells with or without raf1 overexpression. Results of H3K9me3 ChIP-seq were confirmed by two biological replicates in raf1R576H ebs1Δ, raf1R576H raf1-oe, raf1R576H raf1R576H-oe, raf1R576H upf1Δ ago1Δ, sap49A175V, sap49A175V raf1R576H, ubc4-1, ubc4-1 raf1-oe, clr3Δ raf1-oe, clr3Δ upf1Δ, h3.2G13D raf1-oe, h3.2G13D upf1Δ, WT at 37C, raf1-oe at 37C, tor1Δ and tor1Δ upf1Δ cells. Two biological replicates were performed for Raf2-myc ChIP-seq in wild type and raf1 overexpressing cells. Swi6 ChIP-seq results were confirmed by two biological replicates in wild type, raf1R576H, raf1R576H upf1Δ and raf1R576H ebs1Δ cells. Three biological replicates were performed for H3K14ub ChIP-seq in wild type. Two biological replicates were performed to confirm results of H3K14ub ChIP-seq in clr3Δ, clr3Δ raf1-oe, clr3Δ upf1Δ, raf1Δ, ubc4-1, ubc4-1 raf1-oe, h3.2G13D and h3.2G13D upf1Δ mutants. For RNA Seq, two biological replicates were performed for wild type, raf1R576H, upf1Δ and raf1R576H upf1Δ mutants. |
|---|---|
| Sequencing depth | ChIP Seq libraries were sequenced to ~1-3million reads.<br>RNA Seq libraries were sequenced to ~20-50million reads. |
| Antibodies | anti-H3K9me3 (Abcam, Cat#ab8898),<br>anti-H3K14ub (from Jiemin Wong, PMID: 41094145),<br>anti-c-Myc 9E10 (Santa Cruz, sc-40),<br>anti-Swi6 (In-house affinity purified). |
| Peak calling parameters | --broad, --nomodel, --extsize 147 -g 12.57e6 |
| Data quality | Reads were quality-trimmed using fastp. QC reports are generated by fastp. |
| Software | fastp v1.0.1(read trimming), BWA v0.7.17 (alignment of reads from ChIPseq), MACS2 v2.2.7.1 (generation of fold enrichment bedgraphs of ChIPseq data), STAR aligner v2.7.11b (alignment of reads from RNAseq and generation of RPM-normalized bedgraphs) |

