## [Peer Review File · Nature]

Stress controls heterochromatin inheritance via histone H3 ubiquitylation

Corresponding Author: Dr Shiv Grewal

Reviewer #1 was unavailable to evaluate the revised manuscript during the second review round. Reviewer #2, whose expertise overlaps with the relevant areas, reviewed the revisions and deemed them sufficient to support publication.

Version 0:

Reviewer comments:

Referee #1

(Remarks to the Author)

In *S. pombe*, the ClrC complex functions as both an H3K9 methyltransferase and an H3K14 E3 ubiquitin ligase. Raf1, the substrate recognition subunit of ClrC, acts as a DCAF of Cul4 E3 ligases. In this manuscript, the authors generated two Raf1 mutations predicted to disrupt interactions with Cul4. One of these mutations, raf1-R576H, emerged as a partial loss-of-function mutation. A subsequent genetic screen identified suppressor mutations of raf1-R576H: Δ upf1 and Δ ebs1, both of which are involved in the nonsense-mediated decay (NMD) pathway.

The authors demonstrate that Δ upf1 increases raf1 mRNA and Raf1 protein levels, and that Raf1 overexpression (OE) can similarly suppress the phenotypes of the raf1-R576H mutation. Through gradient centrifugation, they show that Raf1 overexpression induces Clr4 to form larger complexes and enhances interactions between Clr4 and ClrC component Raf2. They also report altered Raf2 mobility and increased interactions between Clr4 and Raf2 or Swi6 in Δ upf1 and raf1-OE cells. The authors further claim that Δ upf1 and raf1-OE can suppress mutation of HDAC Clr3 and enhance heterochromatin inheritance.

Finally, authors link these findings to environmental responses, showing that Δ upf1 and raf1-OE enhance epigenetic adaptation to caffeine and heat shock. They also show that TOR signaling regulates Raf1 protein levels and that NMD and TOR pathways antagonize each another.

Overall, the manuscript identifies NMD as a regulator of heterochromatin formation through modulation of Raf1 protein levels and suggests that this regulation is linked to environmental adaptation. While the manuscript addresses an interesting topic, it fails to provide sufficient mechanistic insights into the role of NMD and Raf1 in heterochromatin regulation. Specific weaknesses include insufficient exploration of NMD specificity for raf1 transcript, limited mechanistic investigation of TOR signaling on Raf1 protein levels, and an unclear role of Raf1 in ClrC assembly.

Major Concerns

1. Specificity of NMD regulation of Raf1: The authors conclude that NMD regulates heterochromatin primarily by controlling Raf1 protein levels, supported by evidence of increased raf1 mRNA and Raf1 protein levels in Δ upf1 cells. RNA-seq data suggest raf1 as the primary NMD target, with RNA-IP experiments indicating Upf1 binds the raf1 transcript. However, the mechanism for this specificity remains unexplored. Are other transcripts targeted by Upf1 and do they undergo similar degradation? Sequencing RNA from Upf1 RNA-IP could clarify target specificity and strengthen the conclusions.
2. Role of Epe1 in Δ upf1 phenotypes: RNA-seq data reveal a reduction in epe1 expression in Δ upf1 cells, and caffeine treatment is known to affect Epe1 protein levels. Since Δ epe1 is known to suppress Δ clr3 phenotypes and promote caffeine resistance, it is unclear whether Δ upf1 phenotypes result from reduced epe1 expression or increased Raf1 levels. The level of Raf1 overexpression in raf1-OE cells is not quantified. To validate that Δ upf1 phenotypes primarily result from increased Raf1 expression, the authors should compare phenotypes at Raf1 overexpression levels similar to those in Δ upf1 cells.
3. Role of Raf1 ClrC assembly: the claim that increased Raf1 promotes ClrC assembly is at odds with published results. Glycerol gradient centrifugation (Fig. 3a) suggests that most Clr4 exists as a monomer in wild-type cells, which contradicts previous mass spectrometry studies showing Clr4 is predominantly in a complex. Additionally, how increased Raf1 levels promote ClrC assembly is unclear, as Raf1 is considered peripheral to ClrC (Kuscu et al., 2014, PNAS). Recombinant Raf1

could be used in reconstitution experiments to directly test its role in ClrC assembly in vitro.

4. Involvement of H3K14 ubiquitination: Raf1 is expected to be the substrate recognition subunit of Cul4 E3 ligase, therefore the effects of Δ upf1 and Raf1-OE on H3K14ub is an important question. The authors show that recombinant Raf1 has no effect on ClrC-mediated H3K14 ubiquitination in vitro. However, if increased Raf1 levels affect ClrC assembly, adding recombinant Raf1 to pre-existing ClrC may not be a valid experimental design. The effects of Upf1 and Raf1-OE on H3K14 ubiquitination levels in vivo should be examined.

5. [Text redacted]

6. TOR pathway and Raf1 regulation,: The authors did not address how the TOR-Gad8 signaling pathway regulates Raf1 protein levels, leaving a significant gap.

Minor Concerns

- Referring to Raf1 as a "heterochromatin regulatory hub" is an overstatement.
- In Fig. 3b, 3c, and 3f, variations in Clr4 levels in input samples (e.g., reduced Clr4 in raf1-OE and a slight increase in Δ upf1) should be clarified. If reproducible, these differences warrant explanation.
- Experimental details are missing in some instances. For example, it is not clear how raf1 overexpression was achieved (Pnmt1 or Padh1, which one is used for which experiment). The purity of recombinant Raf1 is not shown.

Referee #2

(Remarks to the Author)

H3K9 methylation is critical for heterochromatin formation. In *S. pombe*, the Clr4 complex (ClrC) is involved in siRNA production and H3K9 methylation, and Raf1, a component of ClrC, plays an essential role in coupling H3K9 methylation and siRNA production. In this manuscript, the authors describe a heterochromatin-heritability regulatory hub (HRH) that broadly regulates heterochromatin in *S. pombe*. They observed that Raf1 was regulated by NMD and TOR signaling pathway in response to different stimuli, altering H3K9me3 levels and heterochromatin distribution. This study provides interesting insights into the heterochromatin regulation, particularly in its response to stimuli.

Considering the roles of Raf1 and other ClrC components in heterochromatin formation are well-documented, and stress-induced responses of H3K9 methylation and heterochromatin have been previously reported, to publish this study in Nature, additional mechanistic insights or functional data are required.

Major points:

1. When Raf1 protein level gets upregulated, are all Raf1 proteins incorporated into the ClrC complex? Do some Raf1 proteins exist alone in a ClrC complex independent manner, which may lead to certain ClrC complex independent function?
2. Why NMD and TOR converge on RAF1? A coincidence or with a specific reason? Does any other stress condition affect heterochromatin via RAF1?
3. The NMD pathway was identified to regulate the level of raf1 mRNA, including WT raf and raf mutants. The mRNA level of raf1R576H is lower than that of WT raf (Fig 2a), is this an NMD-dependent event? If so, how does a point mutant affect mRNA stability?
4. How does the NMD system recognizes raf1 mRNA? The authors mentioned that NMD did not influence mRNA levels of other ClrC components, but its impact appears to vary between WT and raf1R576H genetic background (Table S1). A transcriptome analysis for the impact of NMD-deficiency in WT and Raf1R576H strains may help.
5. A recent paper (PMID: 39477922) reported that ClrC mono-ubiquitinates the disorder region of Clr4, promoting Clr4-Swi6 phase separation and the transition from H3K9me2 to H3K9me3. It is worth testing whether Clr4 mono-ubiquitination enhances the integrity of ClrC and/or its association to chromatin.
6. This manuscript needs an alternative title. TOR-related findings appear only in Fig. 5, with limited exploration of their biological implications.

Minor points:

1. The experimental procedure for RNA IP should be detailed in the methods.

Referee #3

(Remarks to the Author)

In this article, Bhatt et al present evidence for the HDAC-independent mechanisms of heterochromatin propagation in *S. pombe* that they term heterochromatin-heritability regulatory hub (HRH). HRH is centred on Raf1. They further provide data substantiating that the regulation of chromatin association by Clr4SUV39H and heterochromatin epigenetic inheritance via Raf1/HRH may be modulated via NMD- and TOR-dependent mechanisms. Finally, the authors show that changes in temperature and caffeine modulate HRH function and chromatin organization by, at least in part, regulating the levels of Raf1.

Overall, it was thought that this article is likely to be of broad impact and importance since it highlights the previously unappreciated mechanisms that underpin regulation of organization and self-propagation of chromatin by environmental factors and stress. In general, it was thought that the study was well designed and executed. Notwithstanding these clear strengths of the manuscript, several concerns were observed pertinent to functional relationship between HRH and previously reported mechanisms with similar functions, some methodological issues and insufficient data to support some of the author's conclusions. Specific comments are listed below:

Major concerns:

-The mechanism of how NMD regulates Raf1 mRNA levels was found to be incomplete. For instance, the underpinning PTCs were not identified, the increased sensitivity of the mutant raf1 mRNA to NMD remains unclear, etc. The experiments wherein these PTCs are mutated in Raf1 may be warranted as NMD is likely to target a plethora of mRNAs that may also be involved in ensuing phenotypes.

-Related to the above, it is somewhat unclear how the overexpressed Raf1 R576H mutant overcomes NMD.

-Caffeine affects signaling via AKT/TOR axis, which may confound conclusions drawn from the experiments where the effects of caffeine were blamed solely on NMD. Since caffeine appears to bolster Raf1 levels, it would be important to compare the levels of induction of Raf1 with caffeine and those upon overexpression. Conceptually, it may also be helpful to clarify why the overexpression of Raf1 is beneficial under conditions wherein NMD is already inhibited.

-The mechanism linking Tor1 and regulation of Raf1 levels is largely missing. Is this via direct phosphorylation of Raf1, translation, etc.? Also, some potential issues were observed pertinent to interpretation of TOR signaling, as albeit AKT is downstream of TORC2, it is upstream of TORC1, and as far as I know Tor1 is a component of TORC1 and not TORC2 (please see minor comment below). The reported observations can also be explained via modulation of AKT/TORC1 axis at 30°C vs. 37°C. The authors may want to clarify this.

-A fair number of conclusions are drawn from the experiments using reporters. Additional experiments on endogenous genes should perhaps be considered.

-Notwithstanding the similarities between ClrC components and corresponding human proteins, claims of the importance of the described mechanism in cancer should either be experimentally substantiated or toned down.

Minor comments:

-It remains somewhat unclear what is the functional relationship of herein proposed mechanism compared to those that were previously reported and appear to fulfil similar functions (PMID: 32908306, 35879419, 33574613). The authors should consider addressing this throughout the manuscript.

-Were there other genes that were pulled out of the genetic screen in figure 1? If so, the full list of genes should be listed.

-The number of replicates should be indicated in figure legends.

-Cdc2 western blot in figure 5i is of a rather poor quality and hard to interpret. It also appears not to correspond to associated FLAG-Raf1 blot, but this could be due to the poor quality as it is hard to distinguish the borders between the individual lanes.

- The observation that combined upf1/tor1 deletion decreases Raf1 levels as compared to upf1 deletion alone, may not necessarily mean that Upf1 and Tor1 act independently, as e.g., Upf1 may act upstream of Tor1, whereby Tor1 may not be the only target of Upf1 mediating the effects of Raf1.

-..” Among the signaling components tested, the loss of Tor1, a subunit of the TORC2 complex⁶³⁻⁶⁵, caused a severe reduction in Raf1 (Extended Data Fig. 9b, c). In contrast, mutating other kinases, such as the TORC1 subunit Tor2...” Although this may be different in *S. pombe*, I think that the work of Hall and others in *S. cerevisiae* showed that Tor1 and Tor2 are a part of TORC1 and that Tor2 is involved in TORC2. To this end, the authors may want to verify the statement that Tor1 is a component of TORC2.

I hope that the authors find my comments constructive and of sufficient pathos

Sincerely

I/Topisirovic

Version 1:

Reviewer comments:

Referee #2

(Remarks to the Author)
I have no further suggestions.

Referee #3

(Remarks to the Author)
The authors have addressed my concerns in a satisfactory manner. To this end, I have no further comments.

Sincerely

I/Topisirovic

Response to reviewers' comments:

We thank the reviewers for their valuable feedback. Guided by their insightful comments and the editorial recommendations, we have extensively revised the manuscript by conducting additional experiments and updating the text as recommended. These revisions address all reviewer concerns and further strengthen the main conclusions of our study.

Some of the key changes that we have made are as follows:

- (a) An important finding in our paper is that Raf1^{DDb2}, which is believed to serve as the DCAF substrate receptor for the ClrC ubiquitin E3 ligase, is a dosage-critical factor affecting heterochromatin propagation. We have performed additional experiments to address a key question: How do changes in Raf1 levels affect the spreading and epigenetic inheritance of heterochromatin? Our *in vivo* analyses, included in the paper, show that altering Raf1 levels modulates ubiquitylation of histone H3 lysine 14 (H3K14ub), which we demonstrate for the first time is distributed throughout H3K9me3-coated heterochromatin domains. We provide further evidence that H3K14ub is specifically required for the propagation of heterochromatin. In light of recent data from our collaborators showing that H3K14ub is a conserved feature of heterochromatin in human cells (Huang et al., preprint; doi: 10.21203/rs.3.rs-5496475/v1), these findings have major implications for understanding pathways governing the stability of heterochromatin domains in other organisms.
- (b) Our results show that an increase in Raf1 enhances the association of Clr4^{SUV39H} with other ClrC components, although the exact mechanism was previously unclear. We now provide direct evidence that the WD domain-containing region of Raf1 interacts directly with Clr4^{SUV39H}. Moreover, *in vivo*, Clr4^{SUV39H} association with ClrC is proportional to Raf1 levels: Clr4^{SUV39H} fails to associate with ClrC in cells lacking Raf1, and only a weak association is detected when Raf1 levels are low. Consistently, the Clr4 interaction with other ClrC components increases with higher Raf1 levels. These findings support the conclusion that Raf1 serves as a dosage critical factor for the incorporation of Clr4^{SUV39H} into ClrC.
- (c) The enhanced propagation of heterochromatin upon Raf1 upregulation resembles the phenotypes observed upon loss of the anti-silencing factor Epe1. Because Epe1 undergoes ubiquitin-dependent proteolysis, the reviewer asked whether increased Raf1 might reduce Epe1 levels and thereby promote heterochromatin propagation. We addressed this by performing additional experiments: Raf1 upregulation does not alter Epe1 abundance, and our genetic analyses show convincingly that Epe1 and Raf1 act independently to control heterochromatic silencing. Thus, the enhanced heterochromatin propagation caused by increased Raf1 is not due to reduced Epe1 levels.

- (d) Our analyses show that the nonsense-mediated decay (NMD) machinery influences heterochromatin propagation by regulating *raf1* expression. To probe the mechanism, we isolated and characterized two additional mutants using the same screen that initially identified NMD as a regulator of *raf1* expression. One carried a mutation in *upf2*, which functions in NMD; the other harbored a mutation in the *sap49* gene, encoding a U2 snRNP-associated RNA-binding splicing factor. Notably, we found that *raf1* contains inefficiently spliced cryptic introns linked to RNA decay. Consistent with reports that cryptic introns are revealed when RNA-decay pathways are disabled, *raf1* cryptic introns were detected in *upf1* Δ cells. In *sap49* mutant cells, *raf1* mRNA and Raf1 protein are markedly upregulated, mirroring the effect seen in NMD-defective cells. Moreover, increased Raf1 levels in *sap49* mutant cells correlate with enhanced heterochromatin propagation. Taken together with studies implicating splicing factors in stimulating NMD (Aznarez et al., PMID: 29768215; Wen and Brogna, PMID: 20360683), these results suggest that the Sap49 splicing factor may mediate *raf1* mRNA decay via NMD.
- (e) We have substantially revised the paper to incorporate these new results and to address the reviewers' concerns. We have also modified the title to deemphasize TOR and revised the main text in line with the reviewers' suggestions.

The specific comments made by the reviewers, which are indicated in italics, are addressed as follows (responses shown in blue):

Reviewer #1:

The reviewer noted, "*manuscript addresses an interesting topic*" and provided several suggestions to further improve the quality of our study. We sincerely appreciate the insightful suggestions by the reviewer. We have performed additional experiments and addressed all concerns as follows.

1. Specificity of NMD regulation of Raf1: The authors conclude that NMD regulates heterochromatin primarily by controlling Raf1 protein levels, supported by evidence of increased raf1 mRNA and Raf1 protein levels in $\Delta upf1$ cells. RNA-seq data suggest raf1 as the primary NMD target, with RNA-IP experiments indicating Upf1 binds the raf1 transcript. However, the mechanism for this specificity remains unexplored. Are other transcripts targeted by Upf1 and do they undergo similar degradation? Sequencing RNA from Upf1 RNA-IP could clarify target specificity and strengthen the conclusions.

We appreciate the reviewer's comment. NMD is traditionally known for degrading mRNAs with premature termination codons (PTCs); however, growing evidence suggests that it also plays a broader role in gene regulation, including targeting transcripts that lack PTCs (Karam et al., PMID: 23500037). The precise mechanisms that determine the broader range of NMD targets remain to be fully elucidated. Specifically,

how transcripts without PTCs are recognized and targeted by the NMD machinery remains a major question in the field.

Beyond PTCs, other factors reported to influence NMD targeting include the presence of upstream open reading frames (uORFs) adjacent to main ORFs and mRNAs with long 3' UTRs. To this end, we did not detect any uORF upstream of the *raf1* ORF.

Additionally, the *raf1* 3' UTR is 539 bp, which is within the average 3' UTR length in the *S. pombe* genome. To investigate whether the 3' UTR contributes to NMD targeting of *raf1* mRNA, we disrupted the native 3' UTR by inserting a kanamycin resistance marker immediately after the stop codon. When *upf1Δ* was introduced into this strain with the altered *raf1* 3' UTR, we observed a substantial increase in Raf1 levels (**Response Data Fig. 1**→), indicating that alteration of the 3' UTR alone is not sufficient to block NMD-mediated control of *raf1* expression.

Response Data Fig. 1: Disruption of the native 3' UTR does not change the susceptibility of the *raf1* transcript to NMD. FLAG-tagged Raf1 levels were measured by Western blot in the indicated cells carrying a kanamycin resistance marker inserted immediately after the *raf1* stop codon. Ponceau S staining and Cdc2 are shown as a loading control.

We next explored whether secondary structures in the *raf1* transcript influence its targeting by the NMD machinery. The presence of secondary structures may cause ribosome stalling, thereby activating the NMD pathway. Using the RNAstructure platform, we predicted potential secondary structures within the *raf1* mRNA. These analyses indicated that the 330 bases at the 3' end of the *raf1* coding sequence have a high propensity for forming secondary structures (**Response Data Fig. 2**→). To test the effect of these structures, we disrupted the predicted structural potential by introducing synonymous codon mutations in this region. However, these changes did not result in increased *raf1* mRNA levels, nor did they mimic the *upf1Δ* phenotype in suppressing silencing defects in cells with impaired heterochromatin propagation (**Response Data Fig. 2**).

Response Data Fig. 2: Altering the predicted secondary structure of the *raf1* mRNA does not phenocopy *upf1Δ* in suppressing silencing defects in cells with impaired heterochromatin propagation. The modified *raf1* allele fails to suppress heterochromatin silencing defects in cells expressing the histone H3 mutant *hht2-G13D*, whereas loss of Upf1 suppresses these defects (Extended Data Fig. 7).

Analyses of RNA-seq data revealed that *raf1* mRNA contains inefficiently spliced, cryptic introns linked to RNA decay (see **Extended Data Fig. 3a**). Consistent with previous findings that cryptic introns are revealed when pathways involved in target RNA decay are disabled (Lee et al., PMID: 24210919), *raf1* cryptic introns are detected in *upf1Δ* cells (**Extended Data Fig. 3a**). We therefore asked whether splicing machinery promotes *raf1* mRNA decay, as NMD does. An important insight came from characterizing two additional

mutants from the screen that initially identified NMD as a regulator of *raf1* expression (Fig. 1a). One carried a mutation in *upf2* that is also involved in NMD; the other contained a mutation in *sap49*, which encodes a U2 snRNP-associated RNA-binding splicing protein (Extended Data Fig. 3b). Introducing the *sap49*^{A175V} mutation restored heterochromatic silencing and H3K9me3 spreading at the silent *mat* region in *raf1*^{R576H} cells (Extended Data Fig. 3b-d). Notably, the *sap49*^{A175V} mutation significantly upregulated *raf1* mRNA and Raf1 protein, mirroring the effect in NMD-defective *upf1*Δ cells (Extended Data Fig. 3f, g). Together with previous studies implicating splicing factors in stimulating NMD (Aznarez et al., PMID: 29768215; Wen and Brogna, PMID: 20360683), these results suggest that Sap49 may mediate *raf1* mRNA decay via NMD.

2. Role of *Epe1* in $\Delta upf1$ phenotypes: RNA-seq data reveal a reduction in *epe1* expression in $\Delta upf1$ cells, and caffeine treatment is known to affect *Epe1* protein levels. Since $\Delta epe1$ is known to suppress $\Delta clr3$ phenotypes and promote caffeine resistance, it is unclear whether $\Delta upf1$ phenotypes result from reduced *epe1* expression or increased Raf1 levels. The level of Raf1 overexpression in *raf1*-OE cells is not quantified. To validate that $\Delta upf1$ phenotypes primarily result from increased Raf1 expression, the authors should compare phenotypes at Raf1 overexpression levels similar to those in $\Delta upf1$ cells.

We thank the reviewer for raising this important point. The reviewer is concerned that the phenotypes observed in *upf1*Δ cells might be due to the reduction in *Epe1* protein, since *epe1*Δ is known to suppress silencing defects observed in *clr3*Δ cells, rather than the overexpression of Raf1. The reviewer also recommended that we validate the effects of Raf1 overexpression under conditions where Raf1 protein levels are comparable to levels observed in *upf1*Δ cells. We have addressed these points by performing additional experiments, as suggested by the reviewer.

To evaluate the contribution of *Epe1* to the $\Delta upf1$ phenotype, we first examined the transcript levels of *epe1* in $\Delta upf1$. No drastic reduction in *epe1* expression in *upf1*Δ cells was observed when compared to WT cells (Response Data Fig. 3→). However, to more directly address reviewer's concern, we employed a single cell silencing assay using the *GFP* reporter system (*REII*Δ *mat2P::GFP*). In this system, heterochromatin silencing is assayed using either *mat2P* expression that results in dark iodine staining of colonies, or by measuring the number of cells showing green-fluorescent signal due to *mat2P::GFP* expression. Deletion of either *epe1* or *upf1* partially rescued the silencing defect of the *clr3* mutant cells, as indicated by the reduction in iodine staining and restoration of *GFP* reporter silencing in double mutant cells, compared to the *clr3* single mutant (see Extended Data Fig. 6b). Notably, however, the $\Delta epe1$ $\Delta upf1$ double deletion strain exhibited an additive increase in silencing, as indicated by a complete rescue of the silencing defect in *clr3* mutant background cells (Extended Data Fig. 6b). These results, combined with

Response Data Fig. 3: Loss of Upf1 has no major impact on *epe1* expression. RNA-seq shows that *upf1*Δ does not substantially alter *epe1* gene expression.

our results showing that increase in Raf1 levels does not affect Epe1 levels (**Extended Data Fig. 6a**), clearly show that Epe1 and the NMD pathway contribute to silencing through independent mechanisms.

To further address the reviewer's concern, we optimized our analysis of the effects of *raf1* overexpression by using a condition where Raf1 protein levels are comparable to those observed in *upf1Δ* cells. Specifically, we grew cells harboring *raf1* under the control of thiamine-repressible *nmt1* promoter in rich medium (YEA), such that Raf1 levels are similar to *upf1Δ* cells (**Response Data Fig. 4**→; **Extended Data Fig. 3h**). Under these conditions, we observed efficient rescue of the heterochromatic silencing defect in the *raf1^{R576H}* background cells, as assayed using the *REIIΔ mat2P::ura4⁺* reporter (**Extended Data Fig. 3i**). Together, these results support the conclusion that the phenotypes observed in *upf1Δ* cells are primarily driven by increased Raf1 level.

Response Data Fig. 4: The suppression of heterochromatic silencing observed in *raf1^{R576H} upf1Δ* cells are primarily driven by increased Raf1 levels. (a) FLAG-tagged Raf1 levels were measured by Western blot in the indicated cells cultured in rich YEA medium. Ponceau S staining is shown as a loading control. (b) Raf1 expression at levels similar to observed in *upf1Δ* cells is sufficient to suppress heterochromatin silencing defect in *raf1^{R576H}* mutant background cells.

3. Role of Raf1 ClrC assembly: the claim that increased Raf1 promotes ClrC assembly is at odds with published results. Glycerol gradient centrifugation (Fig. 3a) suggests that most Clr4 exists as a monomer in wild-type cells, which contradicts previous mass spectrometry studies showing Clr4 is predominantly in a complex. Additionally, how increased Raf1 levels promote ClrC assembly is unclear, as Raf1 is considered peripheral to ClrC (Kuscu et al., 2014, PNAS). Recombinant Raf1 could be used in reconstitution experiments to directly test its role in ClrC assembly *in vitro*.

Mass spectrometry data from published studies (e.g. Horn et al., PMID: 16024659; Hong et al., PMID: 17114925) do not clearly define the proportion of Clr4 present in complex versus free form. However, our data (**Fig. 3A**) demonstrate that Clr4 increasingly associates with the ClrC complex when Raf1 is overexpressed. According to the structural model by Kuscu et al. (PMID: 24449894), the WD-repeat domain of Raf1 adopts a β -propeller structure, with its wide side binding to Rik1 and its narrow side recruiting substrates. Based on these findings, we hypothesized that Clr4 is recruited to the ClrC complex via Raf1. To test this, we performed *in vitro* binding assays and found that the WD domain-containing region of Raf1 can directly interact with Clr4 (see **Extended Data Fig. 4a**).

To further investigate whether Raf1 is the limiting factor responsible for recruiting Clr4 to ClrC *in vivo*, we used strains with varying levels of Raf1 protein. Specifically, we assessed Clr4 association with the ClrC subunit Raf2 in cells lacking Raf1 (*raf1Δ*), or in cells that expressed low (*raf1^{R576H}*), moderate (*raf1^{WT}*), or high levels of Raf1 protein (*raf1-OE*). Clr4 failed to associate with Raf2 in cells lacking Raf1, and only weak association was detected in *raf1^{R576H}* mutant cells. As Raf1 levels increased in WT cells, the association

between Clr4 and Raf2 also increased (see Fig. 3d). Notably, overexpression of Raf1 in *raf1-OE* cells resulted in a considerable enhancement of Clr4 association with Raf2 compared to both *raf1^{R576H}* and *raf1^{WT}* cells (see Fig. 3d). These results underscore the critical requirement for Raf1 in the incorporation of Clr4 into the ClrC and suggest that Raf1 indeed is the limiting factor for the assembly of this histone methyltransferase/ubiquitin E3 ligase complex.

4. *Involvement of H3K14 ubiquitination: Raf1 is expected to be the substrate recognition subunit of Cul4 E3 ligase, therefore the effects of Δ upf1 and Raf1-OE on H3K14ub is an important question. The authors show that recombinant Raf1 has no effect on ClrC-mediated H3K14 ubiquitination in vitro. However, if increased Raf1 levels affect ClrC assembly, adding recombinant Raf1 to pre-existing ClrC may not be a valid experimental design. The effects of Upf1 and Raf1-OE on H3K14 ubiquitination levels in vivo should be examined.*

As recommended by the reviewer, we investigated histone H3 lysine 14 (H3K14) ubiquitylation levels *in vivo* using an aliquot of anti-H3K14ub antibody provided to us by Dr. Jiemin Wong's lab. Our ChIP-seq analyses presents the first *in vivo* distribution profile of H3K14ub across the *S. pombe* genome. A remarkable finding is that H3K14ub is enriched throughout constitutive heterochromatin domains, such as at pericentromeric domains, subtelomeric loci, and the silent *mat* region (see Fig. 4a). We further show that H3K14ub is completely abolished in cells lacking Raf1 DCAF (see Fig. 4a). Conversely, overexpression of Raf1 leads to a pronounced increase in H3K14ub and H3K9me3 distribution, particularly at subtelomeric regions, which unlike the silent *mat* region and pericentromeric heterochromatin domains, are not flanked by boundary elements (see Fig. 4b). Based in these results, Raf1 is a dosage-critical factor required for H3K14ub, such that merely increasing its levels can enhance heterochromatin propagation.

We also found that H3K14ub is specifically required for heterochromatin propagation. In *ubc4-1* cells, which carry a mutation in the E2 ubiquitin-conjugating enzyme Ubc4, we observed a dramatic reduction in H3K14ub across heterochromatin domains, such as at the silent *mat* region (Fig. 4c). Moreover, the mutation in Ubc4 provided an opportunity to isolate the contribution of H3K14ub to heterochromatin assembly, since Ubc4 is not a core component of the dual functioning ClrC, which mediates both ubiquitylation and H3K9 methylation. We found that loss of H3K14ub in *ubc4-1* cells correlated with impaired heterochromatin spreading and silencing at the silent *mat* region (Fig. 4d). While H3K9me3 was still established at the *cenH* nucleation site, where RNAi machinery recruits ClrC, it failed to spread to surrounding sequences in H3K14 ubiquitination-deficient *ubc4-1* mutant cells (Fig. 4e). Notably, Raf1 overexpression was unable to restore H3K14ub, H3K9me3 spreading or heterochromatic silencing in *ubc4-1* cells (Fig. 4d, e).

Considering the results above, we wondered whether the suppression of defects in heterochromatin propagation observed in *clr3* HDAC mutants and in cells carrying a mutation in one of the three histone H3 copies (H3^{G13D}) upon Raf1 overexpression is connected to changes in H3K14ub. The G13D mutation in histone H3 caused a major reduction in H3K14ub across the silent *mat* domain, including at the *cenH* nucleation site, which retains substantial H3K9me3 enrichment in H3^{G13D} mutant cells (see Extended Data Fig. 7b, c). Thus, the reduction in H3K14ub in H3^{G13D} mutant cells is associated with defects in

heterochromatin spreading, similar to what is observed in *ubc4-1* mutant cells. Remarkably, deletion of *upf1*, which increases Raf1 levels, led to a rise in H3K14ub and restoration of heterochromatin spreading throughout the silent *mat* region in the H3^{G13D} mutant background (see Extended Data Fig. 7b, c). The *upf1Δ* also suppressed defective H3K14ub at the silent *mat* region in Clr3 HDAC-deficient cells (Fig. 5b). Consistently, overexpression of Raf1 (*raf1-OE*) was sufficient to restore both H3K14ub (Fig. 5b) and H3K9me3 distribution in *clr3* mutant background cells (Fig. 5c). These results demonstrate that increasing Raf1 DCAF abundance restores H3K14ub in H3^{G13D} and *clr3Δ* mutants, and that this is functionally coupled to the spread of heterochromatin from the nucleation site to surrounding sequences.

In vitro studies have suggested that H3K14ub dramatically increases Clr4^{SUV39H} activity by enhancing its affinity for the histone substrate and by inducing a conformational shift (Oya et al., PMID: 31468675; Stirpe et al., PMID: 34524082; Du et al., PMID: 40446033). Moreover, H3K14ub is recognized by the amino-terminal chromodomain-containing region of Clr4^{SUV39H} (Oya et al., PMID: 31468675). To this end, our analyses suggests that in addition to promoting Clr4^{SUV39H} chromatin association, the Raf1 DCAF promotes H3K14 ubiquitylation, which in turn stimulates Clr4 methyltransferase activity. This process helps maintain the high levels of H3K9me3 necessary for heterochromatin propagation through the read-write mechanism.

[Redacted text and image]

[Redacted text]

6. *TOR pathway and Raf1 regulation: The authors did not address how the TOR-Gad8 signaling pathway regulates Raf1 protein levels, leaving a significant gap.*

[Redacted text and image]

Minor Concerns:

- Referring to *Raf1* as a “heterochromatin regulatory hub” is an overstatement.

Based on our results, multiple factors converge to regulate levels of *Raf1* protein, which plays an important role in controlling heterochromatin propagation. In addition to the NMD pathway, we have found that *Tor1* also controls the abundance of *Raf1*. Moreover, changing environmental conditions, such as high temperature and caffeine treatment, affect heterochromatin by modulating *Raf1* levels. Given the convergence of multiple regulatory factors affecting *Raf1* levels to modulate heterochromatin propagation and cellular adaptation to changing growth conditions, we believe it is appropriate to refer to this regulatory node as the “heterochromatin heritability regulatory hub.”

- In Fig. 3b, 3c, and 3f, variations in *Clr4* levels in input samples (e.g., reduced *Clr4* in *raf1-OE* and a slight increase in $\Delta upf1$) should be clarified. If reproducible, these differences warrant explanation.

We appreciate the reviewer’s concern and examined whether *upf1 Δ* or *raf1-OE* cause noticeable changes in *Clr4* protein levels. However, we did not observe any reproducible change in *Clr4* protein in either *upf1 Δ* or *raf1-OE* strains compared to WT cells (**Response Data Fig. 7**→). Nonetheless, we have replaced the representative *Clr4* input blot in **Fig. 3b**.

Response Data Fig. 7: Loss of NMD pathway or overexpression of *Raf1* does not alter protein level of *Clr4*. Western blot analysis is used to assay FLAG-*Clr4* levels. Ponceau S staining and *Cdc2* is shown as a loading control.

- Experimental details are missing in some instances. For example, it is not clear how *raf1* overexpression was achieved (*Pnmt1* or *Padh1*, which one is used for which experiment). The purity of recombinant *Raf1* is not shown.

Experimental details, including the promoter used for *raf1* overexpression, are now included in the figure legends.

Reviewer #2:

The reviewer noted, “study provides interesting insights into the heterochromatin regulation, particularly in its response to stimuli”. We are thankful to the reviewer for their encouraging remarks. We have addressed the specific concerns as follows.

Major points:

1. When *Raf1* protein level gets upregulated, are all *Raf1* proteins incorporated into the ClrC complex? Do some *Raf1* proteins exist alone in a ClrC complex independent manner, which may lead to certain ClrC complex independent function?

Although the possibility that some fraction of *Raf1* exists independent of ClrC cannot be ruled out, it is important to note that *Raf1* functions in heterochromatin assembly as part of ClrC. This complex, in addition to methylating histone H3K9, is also required for ubiquitylation of H3K14. Indeed, heterochromatin assembly is abolished when any of the ClrC subunits are mutated, and, to date, no ClrC-independent role for *Raf1* has been reported. The results presented in this study show that *Raf1* abundance influences ClrC chromatin association (**Fig. 3**) and H3K14ub (**Fig. 4**) to maintain high levels of H3K9me3 essential for Clr4^{Suv39h} read-write activity.

2. Why NMD and TOR converge on *RAF1*? A coincidence or with a specific reason? Does any other stress condition affect heterochromatin via *RAF1*?

The reviewer raises an important point. The convergence of regulatory factors such as NMD and Tor1 on *Raf1* to control heterochromatin stability may not be a coincidence. Changes in *Raf1* levels can influence heterochromatin stability, not only by regulating Clr4^{Suv39} incorporation into ClrC (we now show that *Raf1* directly interacts with Clr4), but also by promoting ubiquitylation of H3K14, which stimulates the methyltransferase activity of Clr4^{Suv39h}. Both events—proper ClrC assembly and chromatin association, as well as H3K14 ubiquitylation—increase H3K9me3 density, which is essential for the Clr4^{Suv39h} read-write activity and stable propagation of heterochromatin. Thus, *Raf1* is well positioned to serve as a regulatory hub, integrating signals from different mechanisms to mediate rapid changes in the epigenetic landscape. Our results show that heat stress and caffeine affect heterochromatin via *Raf1*, and future studies may identify additional conditions that utilize this pathway.

3. The NMD pathway was identified to regulate the level of *raf1* mRNA, including WT *raf* and *raf* mutants. The mRNA level of *raf1*^{R576H} is lower than that of WT *raf* (Fig 2a), is this an NMD-dependent event? If so, how does a point mutant affect mRNA stability?

We appreciate the reviewer's comment. Indeed, our analyses show that NMD targets *raf1* mRNA in cells expressing either *raf1*^{WT} or the *raf1*^{R576H} allele. As suggested by the reviewer, we examined whether the lower *raf1* mRNA levels in *raf1*^{R576H} cells result from preferential NMD targeting of the mutant transcript. If so, loss of NMD should equalize *raf1* mRNA levels between *raf1*^{WT} or the *raf1*^{R576H} backgrounds. Northern blot analyses confirmed that *raf1*^{R576H} mRNA is expressed at lower levels than *raf1*^{WT} mRNA (**see Extended Data Fig. 2g**). Deletion of *upf1* caused a substantial increase in *raf1* mRNA levels in both *raf1*^{WT} or the *raf1*^{R576H} cells; however, *raf1* mRNA remained considerably lower in *raf1*^{R576H} *upf1*Δ cells than in *raf1*^{WT} *upf1*Δ cells (**Extended Data Fig. 2g**). These results indicate that NMD targets both *raf1*^{WT} and the *raf1*^{R576H} transcripts, but an additional NMD-independent mechanism likely contributes to reduced *raf1* expression in *raf1*^{R576H} cells. In

this study, we do not focus on the mechanism of the *raf1*^{R576H} mutant; it was used primarily as a genetic tool to perform a suppressor screen that led us to identify *raf1* mRNA as an NMD target. Our conclusion that *raf1* mRNA is an NMD target is now further supported by the identification of an additional mutation in the NMD pathway: in addition to mutations mapping to *upf1* and *esl1*, we have now identified a mutant allele mapping to a gene encoding the NMD component Upf2 that also suppresses *raf1*^{R576H} silencing defects.

4. How does the NMD system recognize *raf1* mRNA? The authors mentioned that NMD did not influence mRNA levels of other ClrC components, but its impact appears to vary between WT and *raf1*R576H genetic background (Table S1). A transcriptome analysis for the impact of NMD-deficiency in WT and *Raf1*R576H strains may help.

As also noted in our response to a comment by reviewer #1, NMD is traditionally known for degrading mRNAs with premature termination codons (PTCs); however, growing evidence suggests that it plays a broader role in gene regulation, including targeting transcripts that lack PTCs (Karam et al PMID: 23500037). The precise mechanisms that determine the broader range of NMD targets remain to be fully elucidated. Specifically, how transcripts without PTCs are recognized and targeted by the NMD machinery is a major question in the field.

Beyond PTCs, other factors reported to influence NMD targeting include the presence of upstream open reading frames (uORFs) adjacent to main ORFs and mRNAs with long 3' UTRs. To this end, we note that no uORF upstream of the *raf1* ORF was detected. Additionally, the *raf1* 3' UTR is 539 bp, which is within the average 3' UTR length in the *S. pombe* genome. To investigate whether the 3' UTR contributes to NMD targeting of *raf1* mRNA, we disrupted the native 3' UTR by inserting a kanamycin resistance marker immediately after the stop codon. When *upf1*Δ was introduced into this strain with the altered *raf1* 3' UTR, we observed a substantial increase in Raf1 levels (see Response Data Fig. 1 above for the response to Reviewer 1's comment 1), indicating that alteration of the 3' UTR alone is not sufficient to block NMD-mediated control of *raf1* expression.

We next explored whether secondary structures in the *raf1* transcript influence its targeting by the NMD machinery. The presence of secondary structures may cause ribosome stalling, thereby activating the NMD pathway. Using the RNAstructure platform, we predicted potential secondary structures within the *raf1* mRNA. We found that the 330 bases at the 3' end of the *raf1* coding sequence have a high propensity for forming secondary structures. To test the effect of these structures, we disrupted the predicted structural potential by introducing synonymous codon mutations in this region. However, these changes did not result in increased *raf1* mRNA levels, nor did they mimic the *upf1*Δ phenotype in suppressing silencing defects in cells with impaired heterochromatin propagation (see Response Data Fig. 2 above for the response to Reviewer 1's comment 1).

Analyses of RNA-seq data revealed that *raf1* mRNA contains inefficiently spliced, cryptic introns linked to RNA decay (see Extended Data Fig. 3a). Consistent with revelation of cryptic introns when pathways

involved in target RNA decay are disabled (Lee et al., PMID: 24210919), *raf1* cryptic introns are detected in *upf1* Δ cells (**Extended Data Fig. 3a**). We therefore asked whether splicing machinery promotes *raf1* mRNA decay, like NMD. An important insight came from characterizing two additional mutants from the screen that initially identified NMD as a regulator of *raf1* expression (**Fig. 1a**). One carried a mutation in *upf2* that is also involved in NMD; the other contained a mutation in *sap49*, which encodes a U2 snRNP-associated RNA-binding splicing protein (**Extended Data Fig. 3b**). Introducing the *sap49*^{A175V} mutation restored heterochromatic silencing and H3K9me3 spreading at the silent *mat* region in *raf1*^{R576H} cells (**Extended Data Fig. 3b-d**). Notably, the *sap49*^{A175V} mutation significantly upregulated *raf1* mRNA and Raf1 protein, mirroring the effect in NMD-defective *upf1* Δ cells (**Extended Data Fig. 3f, g**). Together with previous studies implicating splicing factors in stimulating NMD (Aznarez et al., PMID: 29768215; Wen and Brogna, PMID: 20360683), these results suggest that Sap49 may mediate *raf1* mRNA decay via NMD.

Also, as suggested by the reviewer, we have analyzed results of transcriptome analyses to determine the impact of NMD-deficiency in WT and *raf1*^{R576H} mutant cells. Based on these analyses, we did not detect significant changes in expression of loci encoding other ClrC components.

5. A recent paper (PMID: 39477922) reported that ClrC mono-ubiquitinates the disorder region of Clr4, promoting Clr4-Swi6 phase separation and the transition from H3K9me2 to H3K9me3. It is worth testing whether Clr4 mono-ubiquitination enhances the integrity of ClrC and/or its association to chromatin.

[Redacted text and image]

6. *This manuscript needs an alternative title. TOR-related findings appear only in Fig. 5, with limited exploration of their biological implications.*

Per the reviewer's suggestion, we have removed the reference to TOR from the title.

Minor points:

1. *The experimental procedure for RNA IP should be detailed in the methods.*

As recommended, we have included the RNA IP protocol in the Methods section.

Reviewer #3:

The reviewer stated, "the study was well designed and executed" and noted that "this article is likely to be of broad impact and importance since it highlights the previously unappreciated mechanisms that underpin regulation of organization and self-propagation of chromatin by environmental factors and stress". We sincerely appreciate the reviewer's positive feedback on our study's approach and design, as well as the insightful suggestions. We have addressed the specific concerns as follows.

Major concerns:

-The mechanism of how NMD regulates Raf1 mRNA levels was found to be incomplete. For instance, the underpinning PTCs were not identified, the increased sensitivity of the mutant raf1 mRNA to NMD remains unclear, etc. The experiments wherein these PTCs are mutated in Raf1 may be warranted as NMD is likely to target a plethora of mRNAs that may also be involved in ensuing phenotypes.

Please see our responses to reviewer #2 (comments 3 and 4), who raised similar points. Although NMD is traditionally known for degrading mRNAs with PTCs, growing evidence indicates a broader role in gene regulation, including targeting transcripts without PTCs (Karam et al., PMID: 23500037). In this context, our results suggest an alternative mechanism involving the splicing machinery that likely mediates processing of the *raf1* transcript. As also noted in response to reviewer #2 (comment 3), the reduced expression of the *raf1*^{R576H} mutant is not due to increased sensitivity of this allele to NMD.

Regarding the concern that NMD likely targets numerous mRNAs that might contribute to the ensuing phenotypes, we provide convincing evidence that the restoration of heterochromatin propagation in *raf1*^{R576H} mutant cells upon loss of NMD components is directly linked to reinstatement of Raf1 protein levels. Specifically, the results in **Figs. 2d–f** and **Extended Data Fig. 3h, i** show that expression of Raf1 at levels

comparable to those observed in *upf1* Δ cells is sufficient to restore heterochromatic silencing in the *raf1*^{R576H} background. If other NMD-affected mRNAs were also driving the phenotypes, expressing *raf1* alone would not be expected to mirror the effect of *upf1* Δ in restoring heterochromatin propagation.

-Related to the above, it is somewhat unclear how the overexpressed Raf1R576H mutant overcomes NMD.

NMD targets both *raf1*^{WT} and *raf1*^{R576H} transcripts but reduces, rather than abolishes, *raf1* expression; *raf1*^{R576H} transcripts remain detectable in NMD-proficient cells, confirming that NMD does not eliminate *raf1* transcripts. Overexpressing *raf1*^{R576H} raises *raf1* mRNA and Raf1 protein to levels sufficient for heterochromatin formation. Our analyses indicate that the silencing defect in *raf1*^{R576H} results from reduced Raf1 abundance rather than an intrinsic loss of protein function (see also our response to reviewer #2, comment 3). Accordingly, overexpression of either wild-type or mutant Raf1 rescues the silencing phenotype.

-Caffeine affects signaling via AKT/TOR axis, which may confound conclusions drawn from the experiments where the effects of caffeine were blamed solely on NMD. Since caffeine appears to bolster Raf1 levels, it would be important to compare the levels of induction of Raf1 with caffeine and those upon overexpression. Conceptually, it may also be helpful to clarify why the overexpression of Raf1 is beneficial under conditions wherein NMD is already inhibited.

We appreciate the reviewer's comment. However, our analyses show that Raf1 levels decrease in Tor1-deficient cells but increase upon caffeine treatment. These opposite effects suggest that caffeine enhances *raf1* expression via a TOR-independent mechanism. For overexpression, we used the inducible *nmt1* and *adh1* promoters, yielding Raf1 levels far exceeding those in caffeine-treated cells. Heterochromatin domain assembly, required for cells to acquire caffeine resistance, occurs stochastically and is sensitive to cellular Raf1 levels. Accordingly, the modest Raf1 increase induced by caffeine enables adaptation in only a subset of cells, whereas robust overexpression is markedly more effective.

-The mechanism linking Tor1 and regulation of Raf1 levels is largely missing. Is this via direct phosphorylation of Raf1, translation, etc.? Also, some potential issues were observed pertinent to interpretation of TOR signaling, as albeit AKT is downstream of TORC2, it is upstream of TORC1, and as far as I know Tor1 is a component of TORC1 and not TORC2 (please see minor comment below). The reported observations can also be explained via modulation of AKT/TORC1 axis at 30oC vs. 37oC. The authors may want to clarify this.

[Redacted text]

[Redacted text]

-A fair number of conclusions are drawn from the experiments using reporters. Additional experiments on endogenous genes should perhaps be considered.

Heterochromatin typically assembles at repeat-rich genomic regions, such as pericentromeric repeats. We focused on the well-characterized heterochromatin domain at the silent *mat* region. This ~20-kb domain includes a repeat element that nucleates heterochromatin and developmentally important mating-type genes maintained in a silent state. To monitor heterochromatin assembly, we measured expression of both reporter constructs (e.g., *ura4⁺* or *GFP*) and endogenous silent *mat* genes. Defects in heterochromatin assembly permit expression of the *mat2P::ura4⁺* reporter and result in inappropriate activation the mating-type loci, triggering haploid meiosis and yielding darkly iodine-stained colonies. Throughout this work, we assessed silencing using both endogenous loci and reporter genes, as is standard practice in the field.

-Notwithstanding the similarities between ClrC components and corresponding human proteins, claims of the importance of the described mechanism in cancer should either be experimentally substantiated or toned down.

Given heterochromatin's central role in phenotypic plasticity and reports that H3K9me3 levels shift in response to stress and oncogenic transformation, we discussed the broader implications of these findings, including possible relevance to cancer. In response to the reviewer's comment, we have removed explicit references to cancer and confined related speculation to the Discussion, where we believe is appropriate.

Minor comments:

-It remains somewhat unclear what is the functional relationship of herein proposed mechanism compared to those that were previously reported and appear to fulfil similar functions (PMID: 32908306, 35879419, 33574613). The authors should consider addressing this throughout the manuscript.

The reviewer cites work from the Allshire lab (PMIDs 32908306 and 35879419) showing that caffeine resistance can result from degradation of the anti-silencing factor Epe1. However, we report a distinct pathway that broadly modulates heterochromatin propagation independently of Epe1. Specifically, we uncover a heterochromatin heritability regulatory hub that depends on the abundance of the dosage critical factor Raf1. We provide evidence showing that Raf1 directly interacts with Clr4^{SUV39H} and is a limiting factor for incorporation of this histone methyltransferase activity into ClrC. Moreover, Raf1 DCAF promotes

ubiquitylation of histone H3K14, which stimulates Clr4^{Suv39h} methyltransferase activity, thereby maintaining high H3K9me3 levels essential for self-propagating heterochromatin by the read-write mechanism. This stable propagation enforces heritable gene repression and enables cellular adaptation to changing growth conditions, including resistance to caffeine and high temperature, without requiring genetic mutations.

Regarding our previous work (PMID 33574613), we showed that Tor2 (a TORC1 subunit in *S. pombe*) targets the MTREC RNA-processing complex, which is specifically required for the assembly of small heterochromatin islands at gametogenic genes but does not affect self-propagating heterochromatin domains such as the silent *mat* locus. In the current study, we describe a different mechanism: Tor1 (a TORC2 subunit) controls expression of Raf1 DCAF, a limiting factor that promotes H3K14 ubiquitylation and sustains high H3K9me3 levels, essential for self-propagation of heterochromatin. As shown in **Extended Data Fig. 9b**, Raf1 levels remains unperturbed in *tor2* mutant cells. Thus, the mechanism reported here is distinct and does not overlap with our prior TORC1–MTREC pathway.

-Were there other genes that were pulled out of the genetic screen in figure 1? If so, the full list of genes should be listed.

We have identified two additional mutants from our genetic screen. Consistent with our finding that mutations in the NMD components Upf1 and Esl1 upregulate *raf1* expression and enhance heterochromatin propagation, our analyses show that a mutation in the *upf2* gene, which encodes another NMD component, similarly suppresses heterochromatin defects in a *raf1*^{R576H} mutant background. In addition, we provide evidence showing that a mutation in splicing factor Sap49, a U2 snRNP-associated RNA-binding splicing protein, causes accumulation of *raf1* mRNA, mirroring the effect in NMD-defective *upf1Δ* cells. Interestingly, introducing a mutation in *sap49* restored heterochromatic silencing and H3K9me3 spreading at the silent *mat* region and subtelomeric regions in *raf1*^{R576H} cells. We have added these results to the paper (see **Extended Data Figure 3**).

-The number of replicates should be indicated in figure legends.

As recommended, we have included this information.

-Cdc2 western blot in figure 5i is of a rather poor quality and hard to interpret. It also appears not to correspond to associated FLAG-Raf1 blot, but this could be due to the poor quality as it is hard to distinguish the borders between the individual lanes.

We have repeated this experiment and have included a better-quality western blot result to address this concern.

- The observation that combined *upf1/tor1* deletion decreases Raf1 levels as compared to *upf1* deletion alone, may not necessarily mean that Upf1 and Tor1 act independently, as e.g., Upf1 may act upstream of Tor1, whereby Tor1 may not be the only target of Upf1 mediating the effects of Raf1.

We appreciate the reviewer's point. RNA immunoprecipitation (RNA-IP) shows that Upf1 directly binds the *raf1* mRNA (Fig. 2b), which is upregulated in cells lacking Upf1 (see also northern blot in Extended Data Fig. 2g). These observations indicate that the NMD pathway acts directly on the *raf1* mRNA to promote its degradation, arguing against Tor1 regulating *raf1* downstream of Upf1. Moreover, we note that Raf1 protein levels decreased upon loss of Tor1 in *upf1Δ* cells (Fig. 6i), suggesting that Tor1 and Upf1 control *raf1* expression through independent mechanisms.

-..” Among the signaling components tested, the loss of Tor1, a subunit of the TORC2 complex⁶³⁻⁶⁵, caused a severe reduction in Raf1 (Extended Data Fig. 9b, c). In contrast, mutating other kinases, such as the TORC1 subunit Tor2...” Although this may be different in *S. pombe*, I think that the work of Hall and others in *S. cerevisiae* showed that Tor1 and Tor2 are a part of TORC1 and that Tor2 is involved in TORC2. To this end, the authors may want to verify the statement that Tor1 is a component of TORC2.

The nomenclature is reversed between *S. cerevisiae* and *S. pombe*: in *S. cerevisiae*, Tor1 is a TORC1 subunit and Tor2 is a TORC2 subunit, whereas in *S. pombe*, Tor1 is part of TORC2 and Tor2 is part of TORC1.

We believe that the revised manuscript now meets the standards of quality and novelty expected for publication in *Nature*.

Response to reviewers' comments:

We sincerely thank the reviewers for their valuable feedback and for supporting the publication of our work.